



# The differences between remote sensing and in situ air pollutants measurements over the Canadian Oil Sands

Xiaoyi Zhao[1], Vitali Fioletov[1], Debora Griffin[1], Chris McLinden[1], Ralf Staebler[1], Cristian Mihele[1], Kevin Strawbridge[1], Jonathan Davies[1], Ihab Abboud[1], Sum Chi Lee[1], Alexander Cede[2,3], Martin Tiefengraber[3,4], Robert Swap[2]

[1]Air Quality Research Division, Environment and Climate Change Canada, Toronto, ON, Canada
[2]NASA Goddard Space Flight Center, Greenbelt, MD 20771, USA.
[3]LuftBlick, Innsbruck, Austria.
[4]Department of Atmospheric and Cryospheric Sciences, University of Innsbruck, Innsbruck, Austria.

*Correspondence to*: Xiaoyi Zhao and Vitali Fioletov (xiaoyi.zhao@ec.gc.ca and vitali.fioletov@ec.gc.ca)

**Abstract**. Ground-based remote sensing instruments have been widely used for atmospheric research but applications for air quality monitoring remain limited. Compared to an in situ instrument that provides air quality conditions at the ground level, most remote sensing instruments are sensitive to a broad range of altitudes, often providing only integrated column observations. These column data can be more difficult to interpret and to relate to surface values and hence to "nose-height-level" health factors. This research utilized ground-based remote sensing and in situ air quality observations in the Canadian Oil Sands Region to investigate some of their differences.

Vertical column densities (VCDs) of $SO_2$ and $NO_2$ retrieved by Pandora spectrometers located at the Oski-Otin site at Fort McKay, (Alberta, Canada), from 2013–2019 were analyzed along with measurements of $SO_2$ and $NO_2$ surface concentrations and meteorological data. Aerosol optical depth (AOD) observations by CIMEL sunphotometer were compared with surface $PM_{2.5}$ data. The Oski-Otin site is surrounded by several large bitumen mining operations within the Athabasca Oil Sands Region (AOSR) with significant $NO_2$ emissions from the mining fleet. Two major bitumen upgraders that are 20 km south-east of the site have total $SO_2$ and $NO_2$ emissions of about 40 kt yr$^{-1}$ and 20 kt yr$^{-1}$ respectively. It was demonstrated that remote sensing data from Pandora and CIMEL combined with high vertical resolution wind profiles can provide information about pollution sources and plume characteristics. Elevated $SO_2$ VCDs are clearly observed for times with south and south-eastern winds, particularly at 200–300 m altitude (above ground level). High $NO_2$ VCD values were observed from other directions (e.g., north-west) with less prominent impacts from 200–300 m winds. In situ ground observations of $SO_2$ and $NO_2$ show a different sensitivity with wind profiles, indicating they are less sensitive to elevated plumes than remote sensing instruments. In addition to measured wind data and lidar observed boundary layer height (BLH), modelled wind profiles and BLH from ERA-5 have been used to further examine the correlation between column and surface observations. The results



show that the ratio of measured column and surface concentration values could show positive or negative correlation with BLH, which depends on the height of emission sources (e.g., emissions from high stacks or near surface).

This study explores differences between remote sensing and in situ instruments in terms of their vertical, horizontal, and temporal sampling differences. Understanding and resolving these differences are critical for future analyses linking
satellite, ground-based remote sensing, and in situ observations in air quality monitoring and research.

## 1. Introduction

Satellite measurements have been widely used to analyze long-term changes and trends in atmospheric air pollutants such as sulphur dioxide ($SO_2$) and nitrogen dioxide ($NO_2$) (Song and Yang, 2014; Wang et al., 2015; McLinden et al., 2016; Duncan et al., 2016; Krotkov et al., 2016; Barkley et al., 2017; Liu et al., 2017) and to estimate the corresponding emission rates
(Streets et al., 2013; Fioletov et al., 2011, 2015; Lu et al., 2015; de Foy et al., 2015). It is expected that satellite instruments will play an even bigger role in air quality monitoring in the future (new high-resolution geostationary missions such as TEMPO (Zoogman et al., 2017), Sentinel 4 (Gulde et al., 2017) and GEMS (Kim et al., 2020)). However, there is a major complication in applying satellite measurements to surface air quality applications. Most nadir viewing satellite instruments are sensitive to the entire atmospheric column and hence derive vertical column densities (VCD; vertically integrated number
density of a given species from the bottom to top of the atmosphere) and their conversion to surface concentrations required by air quality applications is not straightforward.

Ground-based remote sensing observations of the same quantity help both to validate satellite observations and facilitate a better interpretation of satellite data and their links to surface concentration (Richter et al., 2013), keeping in mind the mismatch between satellite and ground-based remote sensing quantities that was widely reported (Herman et al., 2009;
Kollonige et al., 2017; Judd et al., 2020; Zhao et al., 2020, 2022). Alternatively, a direct comparison of ground-based VCD observations with surface concentrations does not typically produce high correlations even in polluted urban areas (Dieudonné et al., 2013). Taking the planetary boundary layer (PBL) height into account improves the agreement between in situ and VCD-derived surface mixing ratios for $NO_2$, but not so much for $SO_2$ (Knepp et al., 2015). Other efforts to convert ground-based VCD measurements to surface concentrations were made by researchers via various approaches, including applying convertion
ratio factors (e.g., Kollonige et al., 2017; Zhao et al., 2019). However, due to some fundamental differences between ground-based remote sensing and in situ instruments (e.g., sampling difference in space and frequency), surface concentration estimates via these instruments still have non-negligible biases difference (e.g., surface $NO_2$ derived from Pandora observations has about -7% bias compared to in situ data; Zhao et al., 2019). One of the major challenges is the difficulty in modelling, measuring, or estimating the boundary layer conditions (e.g., boundary layer height (BLH), wind speed and direction profiles)
that is critical to resolving the vertical structure of air pollutants (e.g., Zhao et al., 2019).





The mining and refinery operations in the Athabasca Oil Sand Region (AOSR) are one of the largest sources of atmospheric pollutants in Canada. Environmental and health concerns associated with oil sands operations, including air quality and acid deposition, are well known (e.g., Kelly et al., 2010). $SO_2$ and $NO_2$ are on the top of the list of gaseous pollutants emitted from the oil sands and highly elevated levels of these pollutants over the oil sands area have been detected (Simpson et al., 2010). Due to the large size of the oil sands operation area, satellite column measurements are an attractive method for air pollution monitoring in this region and satellites have been used for monitoring $SO_2$ and $NO_2$ emissions and trends in AOSR (McLinden et al., 2012, 2014, 2016, 2020). The oil sands operation activities north and south of the Oski-Otin site at Fort McKay in AOSR make the pollution conditions highly dependent on wind direction on certain altitudes.

To better establish the link between column and surface concentrations observations, we examined seven years of ground-based remote sensing column observations along with surface concentration measurements, vertical wind profile and BLH observations, and reanalysis model data at the Oski-Otin site. In this work, vertical column of $SO_2$ and $NO_2$ are provided by Pandora spectrometer (Herman et al., 2009; Fioletov et al., 2016); aerosol optical depth (AOD) are provided by CIMEL sunphotometer (Holben et al., 2001; Sioris et al., 2017); $SO_2$, $NO_2$, and $PM_{2.5}$ surface concentration values are from Thermo 43i, 42i, and Teledyne API T640, respectively. Collocated wind profiler and lidar instrumentation (Strawbridge et al., 2018) allow for examining the VCD dependence on the wind speed and direction at different altitudes and linking the VCDs to surface concentrations ratio with observed BLH. Besides trace gas pollutants, we also compare and study the differences between surface $PM_{2.5}$ observations with remote sensing AOD data. $PM_{2.5}$ concentration is also one of the three indicators in the Canadian Air Quality Health Index (Stieb et al., 2008). Remote sensing observations of aerosols face more challenges than ozone and $NO_2$.

Utilizing information on detailed vertical wind field profiles and BLH, this work illustrates the cause and demonstrates the differences between ground-based remote sensing and in situ observations, as well as studies the ratios of the surface to column values for various wind directions and BLH values. This paper is organized as follows: Sect. 2 describes the ground-based remote sensing observations and in situ measurements. In Sect. 3, the vertical sampling difference between remote sensing and in situ measurements are evaluated by integrating the data with wind profiles for $SO_2$, $NO_2$, and aerosols respectively. Horizontal transports of pollutants are evaluated with wind speed and directions in Sect. 4 to understand their transport patterns. Sect. 4 also focuses on how meteorological conditions can affect the sampling differences between remote sensing observations and in situ measurements. Sect. 5 shows the potential of using reanalysis meteorological data to help interpret the ground-based remote sensing observations. In Sect. 6, the integration period differences are evaluated. Conclusions are given in Sect. 7.



## 2. Observation sites and datasets

The Oski-Otin site at Fort McKay (57.184°N, 111.64°W) is equipped with various instruments for air quality measurements (e.g., Strawbridge, 2013; Fioletov et al., 2016; McLinden et al., 2020). Fort McKay is a small town (population of 600) surrounded by seven oil sands surface and two in-situ mining facilities to the north and south. There are two major $SO_2$ and

$NO_2$ sources located south of Fort McKay: The Syncrude Mildred Lake plant is located 16 km to the south of Fort McKay and the Suncor Millenium Plant is 23 km south-south-east. According to the National Pollutant Release Inventory (NPRI, https://www.canada.ca/en/services/environment/pollution-waste-management/national-pollutant-release-inventory.html, last accessed on Sept. 10, 2023), the 2013–2019 annual mean emissions were about 26 kt (14 kt) and 17 kt (5.8 kt) of $SO_2$ ($NO_2$) per year from the Syncrude and Suncor facilities, respectively. McLinden et al. (2020) reported discrepancies in NPRI and the

satellite-derived $SO_2$ emissions in AOSR, peaking at 50 kt yr$^{-1}$ around 2016 (i.e., NPRI underestimated the emissions; Pandora instrument show agreement with satellite observations). There is also the Horizon Oil Sands processing plant and mine 18 km to the north, but the emissions from that source are smaller (4 kt of $SO_2$ and 1.4 kt of $NO_2$ per year). For $NO_2$, there are however many other small local sources. In addition, there are major $NO_2$ emissions from local sources as well as from the mining areas caused by the off-road heavy vehicle fleet that excavates and transports bitumen from the mines to the on-site separation

facilities. There are hundreds of kilometres of pristine boreal forests to the west and east from Fort McKay with no $SO_2$ and $NO_2$ sources. Thus, the pollution level at Fort McKay is largely dependent on the wind direction. In situ data from Fort McKay can be found from the Wood Buffalo Environmental Association (WBEA) (site AMS 01; https://wbea.org/ last accessed on Sept. 10, 2023)

### 2.1 Pandora

The Pandora spectrometer is a recently developed instrument that measures solar and sky spectral radiation in the UV and visible part of the spectrum (Herman et al., 2009; Szykman et al., 2019). Direct sun (DS) measurements are the main type of observations, although it is also capable of operating in the zenith sky (ZS) and multi-axis differential optical absorption spectroscopy (MAX-DOAS) modes (e.g., Zhao et al., 2019; Kreher et al., 2020). In this work, vertical column densities of trace gases are derived from the DS measured spectra. It has been demonstrated that Pandora can successfully observe total

column ozone and $NO_2$ (Herman et al., 2009, 2015; Tzortziou et al., 2012, 2015; Zhao et al., 2016, 2020). In 2013, a Pandora instrument, along with other instrumentation, was deployed to the AOSR and its measurements were used to establish optimal retrieval procedures for the $SO_2$ VCD measurements and estimate measurement uncertainties (Fioletov et al., 2016).

The Pandora instrument consists of an optical head sensor, mounted on a computer-controlled sun-tracker, and connected to a commercial Avantes array spectrometer by means of an optical fibre. To allow for the detection of different

absorbers, the instrument periodically measures UV spectra with the U340 bandpass filter with a cut-off limit at 380 nm on and off, with an interval of about 90 seconds. The 306–330 nm spectral interval was used for $SO_2$ spectral retrievals (ECCC



research retrieval) and the 400 to 440 nm interval was used to retrieve $NO_2$ (PGN official retrieval, version nvs1p1-7) (Fioletov et al., 2016; Zhao et al., 2020). The integrated vertical column values of trace gases are reported in Dobson Units (DU; DU = $2.69\times10^{16}$ molecules cm$^{-2}$). Pandora instrument no. 104 was deployed to Fort McKay in the oil sands region, where it was operating from August 15, 2013 to November 14, 2013. It was then redeployed on August 21, 2014, and was operational until late 2015 (Zhao et al., 2016). From September 2017 to 2020, Pandora no. 122 was deployed at the site. The Pandora no. 122 was operated with multi-axis observations (in addition to DS and ZS) since 2018, but no $SO_2$ or $NO_2$ profile has been retrieved from these observations yet. The operation of Pandora at the site stopped in summer 2020.

A detailed description of the Pandora spectrometer and its total column $NO_2$ retrieval algorithm is given by (Herman et al., 2009). In order to isolate tropospheric $NO_2$ VCD from the total column VCD measured by Pandora, stratospheric $NO_2$ partial columns were subtracted from Pandora measurements (using OMI satellite data and the Pratmo box model; Zhao et al., 2019). For the observation period in the oil sands region, typically, stratospheric $NO_2$ accounts for about 46% of the total column (median value), with a standard deviation of 35%. For convenience, we refer to this tropospheric $NO_2$ VCD as simple "$NO_2$ VCD". Information about the instrument set up at Fort McKay and the $SO_2$ data and algorithms are available from Fioletov et al., (2016). For $SO_2$ data, as the only sources are near the surface (as no comparable $SO_2$ quantities were in the stratosphere during the analyzed period), the retrieved total column $SO_2$ ($SO_2$ VCD) are directly been used in this study.

## 2.2 CIMEL Sunphotometer

CIMEL Sunphotometers for measuring aerosol properties were deployed the Oski-Otin site since 2013, as a part of the AEROCAN network (Sioris et al., 2017). AEROCAN is the sub-network that consists of twenty sites across Canada within the Aerosol Robotic Network (AERONET, https://aeronet.gsfc.nasa.gov/ last access: 10 September 2023) that was established in the early 1990s (Holben et al., 2001). The sunphotometer measures AOD in direct sun mode at eight wavelengths, typically 340, 380, 440, 500, 675, 870, 940, and 1020 nm. AOD is a unitless quantity representing the vertically integrated extinction of radiation due to scattering and absorption of particles. To better quantify fine and coarse mode components of the measured AOD, spectral deconvolution algorithm (SDA) products were developed by O'Neil et al. (2003) based on the AOD spectral dependence and higher order spectral derivatives. Fine and coarse modes are essentially comprised of sub-micron and super-micron particle radii, respectively (O'Neill et al., 2001). In Fort McKay, Sioris et al. (2017) reported that coarse mode aerosol and PM$_{2.5}$ only had low temporal correlation (as -0.02), while fine mode aerosol and PM$_{2.5}$ had much higher correlation (as 0.53). In this work, the fine mode aerosol (SDA Level 2 version 4.1; quality assured) outputs are used, which are processed from an operational product of AERONET.



## 2.3 WindRASS

The Radio-acoustic Sounding System (RASS) wind profiler (WindRASS; model MFAS, Scintec, Rottenburg, Germany) was another instrument installed at the Oski-Otin site (Gordon et al., 2018; Strawbridge et al., 2018). It reports wind speed and direction as well as other meteorological parameters as 15-minute averages, at 77 levels from 40 m to up to 800 m above

ground (10 m vertical resolution). It is a monostatic 64-transducer sodar that emits sound pulses near 2000 Hz vertically and in the four cardinal directions (tilted at 22° and 29° from zenith). The radio antennas then emit electromagnetic waves (915 MHz) that are partially reflected by the sound waves as they propagate away into the atmosphere. Doppler analysis of the returning signal is used to reconstruct the temperature profile (based on the fact that the speed of sound is a function of the square root of the virtual temperature) as well as the 3-D wind vector. Note that the number of successful WindRASS

observations depends on the altitude and the vertical range varies from measurement to measurement as discussed in Appendix A.

## 2.4 Lidar

An autonomous lidar system that can be monitored remotely and operated continuously (except during precipitation events) was also installed at the Oski-Otin site since 2013 (Strawbridge, 2013). The lidar simultaneously emits two wavelengths laser

light (1064 and 532 nm, Q-switch Nd:YAG) at energies of approximately 150 mJ pulse$^{-1}$ wavelength$^{-1}$ and detects the backscatter signal at 1064 nm and both polarizations at 532 nm. Since 2016, the lidar system was upgraded by adding an ozone DIAL (Differential Absorption Lidar) to simultaneously measure the vertical profile of tropospheric ozone, aerosol (at 355, 532, and 1064 nm) and water vapour (through the addition of the 355 nm output channel to the aerosol lidar) from near the ground to 10 to 15 km (Strawbridge et al., 2018). The BLH was derived via the 532 nm aerosol lidar profile. Details on the

BLH processing algorithm can be found from Strawbridge and Snyder (2004).

## 2.5 In situ measurements

In this study, we used the in situ measurements of $SO_2$, $NO_2$, $PM_{2.5}$, NOx, and $O_3$ as well as the wind speed and direction data provided by the Wood Buffalo Environmental Association (WBEA; https://wbea.org/, last accessed on Sept. 10, 2023). The list of analytical equipment that have been used can be found at https://wbea.org/wp-content/uploads/2021/03/Bertha-Ganter-

Fort-McKay-Site-Documentation_2021.pdf (last accessed on Sept. 10, 2023). Surface $SO_2$ and $NO_2$ were measured by Thermo Scientific 43i and 42i instruments with a precision of 1 and 0.4 ppbv, respectively. $PM_{2.5}$ were measured by Teledyne API T640 instrument with a precision of 0.5 µg m$^{-3}$. The Level 2 data (hourly) that has undergone validation review are used in this work. Pandora, sunphotometer, and WindRASS data were averaged into hourly resolution to match with sampling rate of in situ Level 2 data.





## 2.6 ERA-5 Reanalysis meteorological data

In this work, ECMWF Reanalysis v5 (ERA-5) wind data (hourly, using model-levels as the vertical co-ordinate) are utilized in addition to WindRASS observations. Several studies have verified that ERA-5 wind data can facilitate remote-sensing-based emission estimations (e.g., Fioletov et al., 2015; McLinden et al., 2020), regional air quality monitoring (e.g., Liu et al., 2021; Tzortziou et al., 2022; Zhao et al., 2022), and wind-based satellite validation (Zhao et al., 2020; Park et al., 2022). To assess whether ERA-5 wind can be a good replacement for near-surface to free troposphere vertical wind measurements, the ERA-5 model level data (Hersbach et al., 2020) were used (instead of its pressure level data, which is more commonly used). It is worth noting that for ERA-5 pressure level wind data, there are only 5 wind layers from 900 to 1000 hPa (900, 925, 950, 975, and 1000 hPa). While using ERA-5 model level data, there are about 11 to 18 modelled wind layers from 900 to 1000 hPa, for the Oski-Otin site. Thus, ERA-5 model level data can provide a much more detailed vertical wind field than pressure level data. In addition to wind data, the boundary layer height from ERA-5 has also been used to examine the correlation between column and surface observations.

## 3. Vertical sampling differences between remote sensing and in situ observations

Figure 1 shows the time series of remote sensing and in situ observed $SO_2$, $NO_2$, and aerosol data in Fort McKay. Compared to in situ observations that practically have no gaps, remote sensing data records have gaps due to instrumental issues and other operational changes as well as due to the clouds. Thus, only coincident observations from both remote sensing and in situ instruments were included in the analysis.

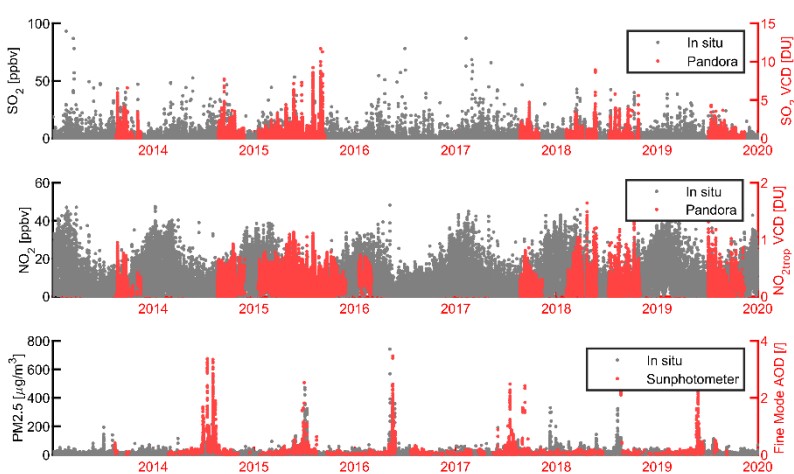

**Figure 1. Time series of remote sensing and in situ observations of SO₂, NO₂, and aerosol data at Fort McKay.**

### 3.1 Sensitivity of SO₂ observations to winds at different levels

For the Pandora located at Oski-Otin site, higher SO₂ VCDs for southern wind direction were previously reported based on

the surface winds (Fioletov et al., 2016). Utilizing SO₂ VCD observations, various satellite measured SO₂ plume reconstructions work (e.g., McLinden et al., 2020) has been done to estimate the emissions (e.g., Fioletov et al., 2015, 2020). Typically, in those plume models, the vertical structure of the SO₂ plume is not considered, as satellites measure the integrated column of the SO₂. However, to link the surface and satellite observations, information of the vertical structure of the SO₂ plume or other pollutants are critical. If the plume is elevated, it is not necessarily true that the direction of surface winds would

be correlated with the VCDs; the wind direction at the plume height would be a more important factor. For the surface in situ instruments, this effect would not be prominent (i.e., in situ instruments are expected to most sensitive to near surface wind). To test that, we calculated mean VCDs for data binned by the wind altitude and direction using the wind profiler data. If the mean SO₂ VCD values are the same for all wind directions, then winds at that height do not affect SO₂ transport. And the larger the spread of the mean SO₂ VCD values for different directions, the greater impact from winds at that height is for SO₂

transport. This information can be used to determine what wind directions and at what altitudes have the largest impact on VCDs and thus for pollutant transports.

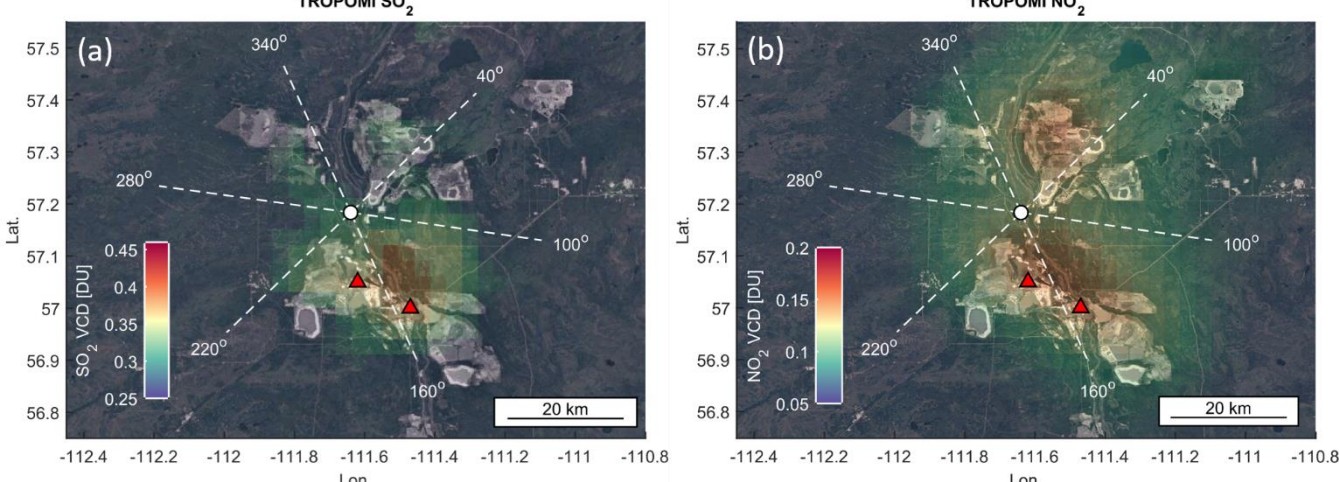

**Figure 2. Satellite maps (© Google Maps) of the Athabasca Oil Sand Region (AOSR) masked with selected wind directions. The Pandora spectrometer, sunphotometer, WindRASS, lidar, and in situ instrument were located at the observation site represented by a white circle. The two largest upgraders in the mining areas are shown by red triangles. The white dashed lines show the centre of the wind sectors. Maps are masked with pixel averaging of total column $SO_2$ and tropospheric column $NO_2$ (2018–2021) from TROPOMI satellite instrument (McLinden et al., 2020).**

A map of the AOSR is shown in Fig. 2, with selected wind directions plotted. The wind direction bins were selected to have one bin centred at 160° where the $SO_2$ signals have a maximum from the two major upgraders (see red triangles on the map). Elevated $SO_2$ plumes from these upgrader stacks were also observed and reported by a MAX-DOAS instrument in previous studies (e.g., Davis et al., 2020). The heights of these stacks are in the range of 76–183 m (Gordon et al., 2018). Davis et al. (2020) reported the retrieved $SO_2$ plume can reach 30 ppbv concentration at 395 m height.

Figure 3a shows the mean $SO_2$ VCD for data binned into 60°-wide wind direction bins based on WindRASS wind data from each 10 m altitude level. In other words, Fig. 3a shows the results of binning measured $SO_2$ VCDs with wind information from different vertical wind layers; Fig. 3a is not depicting actual vertical profiles of $SO_2$, but an illustration of which layer of measured wind profile has the highest impact on $SO_2$ column and surface observations. The values are the highest for winds out of the south-south-easterly (SSE; 160°) at all altitudes. The mean values are close to zero for the 40°, 280°, and 340° sectors. This dependence of VCDs on the direction is very similar to that of surface winds (Fioletov et al., 2016). The magnitude of the mean $SO_2$ VCD in both cold (Nov. to Apr.) and warm (May to Oct.) seasons reaches its maximum of 0.39 DU at 200–300 m; about 38% higher than the mean value for the same direction of surface or 40 m winds (0.25 DU). Thus, in the analyzed altitude range, the wind direction at 200–300 m has the largest impact on Pandora measured $SO_2$ VCD. In contrast, for the clean air directions (280°) $SO_2$ VCD are the lowest regardless of the wind altitude. Also, wind data above 300 m (250 m for the cold season) were too intermittent to separate $SO_2$ VCD signals which indicate the $SO_2$ plumes were within this height (WindRASS vertical data recovery rate is shown in Fig. A1).



Compared to Pandora measured $SO_2$ VCDs, in situ $SO_2$ surface concentrations show a similar but not identical picture (see Fig. 3b). The in situ $SO_2$ observations show a similar increased sensitivity for winds at levels of 200–300 m with the same polluted and clean air directions as 160° and 280° respectively. However, the magnitude of the binned mean in situ $SO_2$ reaches its maximum of 3.16 ppbv for wind levels at 200–300 m is only about 27% higher than the mean value for the same direction at the surface layer winds (2.32 ppbv). The results indicate the different characteristics between the two measurement techniques of $SO_2$, i.e., VCD values are more sensitive to elevated plumes than surface values (i.e., the $SO_2$ emitted from high stacks of refineries in this case; 38% increased VCDs, while only 27% increased for surface concentrations, when using 200–300 m wind). In other words, in conditions that wind at 200–300 m are from 160°, on average, the observation yields the highest $SO_2$ VCDs, even if the plume is not fully vertically mixed to reach the ground level (see more discussion in Sect. 5.2).

As mentioned, the WindRASS wind profiler data have different number of successful observations at different heights. The number of available successful measurements as a function of altitude declines almost linearly. Only about 75% of all wind profiles reach 200 m and only 12% reach 400 m (see Fig. A1). Thus, the vertical structure revealed in Fig. 3 shows increased uncertainties for high altitudes (e.g., see increased width of the 1-sigma envelopes for profiles). This leads to a bias of wind profile observations that reach 500 m of altitude towards low wind speed conditions. Thus, the wind profile data from WindRASS is used only if it has full coverage within 0–300 m for both the warm and cold seasons, thus minimizing the wind speed bias while retaining most sensitivity to to resolve air pollutant transport patterns in this region.

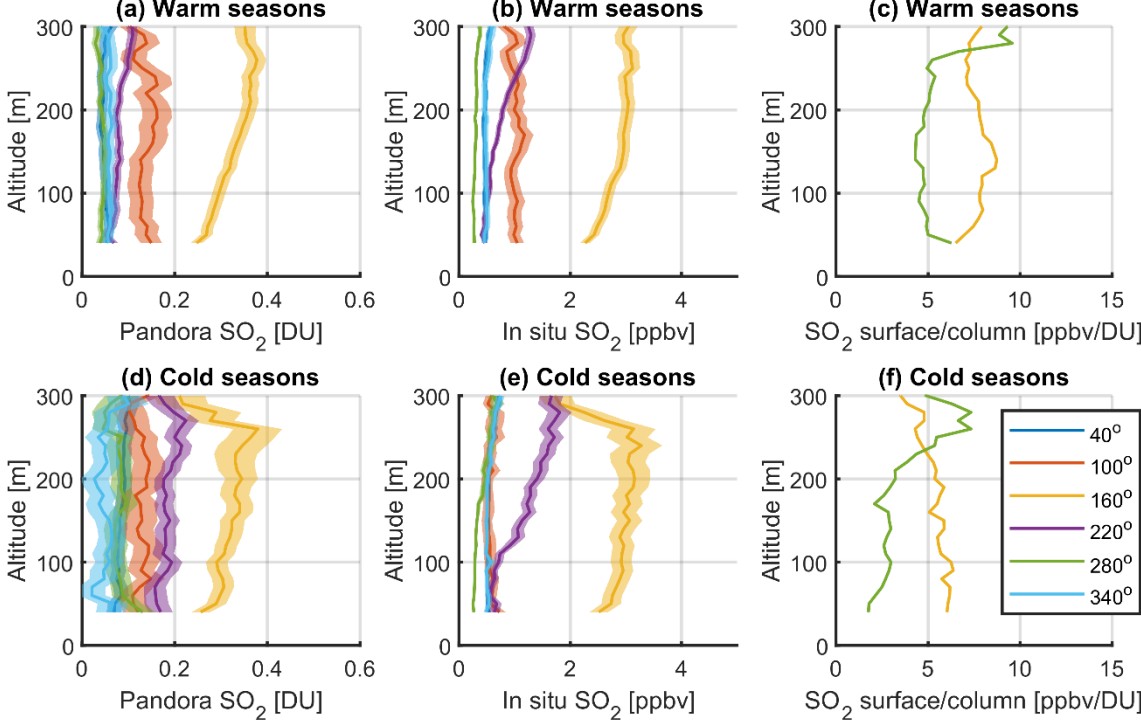

**Figure 3. The mean SO₂ column (a and d), surface (b and e) observations, and their median ratios (c and f) binned by wind**
5    **directions at different altitudes at Oski-Otin. Wind layer information (altitude and direction) was measured by WindRASS, SO₂**
**column observations were from Pandora, surface measurements were from in situ instrument. Color lines show the mean values of**
**SO₂ with 1-sigma envelopes. A sharp maximum can be found when the wind is from 160°, the directions to the upgraders. The wind**
**direction bins were selected to have one bin centred at 160° where the SO₂ signals have a maximum. Data were categorized by warm**
**(May to Oct.; top row) and cold (Nov. to Apr.; bottom row) seasons.**

It is expected that the SO₂ plume height will be affected by boundary layer dynamics that are reflected by temperature

and other seasonal meteorological aspects. Thus, the sensitivity of SO₂ VCD to the wind direction were categorized into warm

(Figs. 3a and b) and cold seasons (Figs. 3d and e). Compared to the graduate decreasing sensitivity above "plume height" in

warm seasons (i.e., 200–300 m), both remote sensing and in situ observations show their sensitivity to SO₂ emissions decreased

15    sharply after passing the plume height in cold seasons. The results may indicate that the vertical transport of SO₂ is more

refined within the boundary layer due to a lack of vertical mixing in cold temperatures. Another interesting factor is the SO₂

plume from 220° is more prominent for in situ data (see the purple line in Fig. 2e). However, there are no known strong SO₂

emission sources from this direction, and this feature is not captured by Pandora.



Figs. 2c and f show the ratio of observed surface concentration to column values (from 160° and 280°) grouped by the wind direction at different altitudes. For $SO_2$, this surface to column ratio preserved the pattern discussed above. Compared to the clean air directions (280°), such ratio values of the directions of the upgraders (160°) are much higher, indicating the $SO_2$ pollutants from the upgraders are less vertically mixed than its background conditions. However, more detailed features can be revealed if the data is further analyzed with BLH information (more details will be discussed in Sect. 5.2).

### 3.2 Sensitivity of $NO_2$ observations to winds at different levels

Analysis of $NO_2$ data is shown in Fig. 4. Similar to $SO_2$ VCD, the mean $NO_2$ VCD (tropospheric column, see description in Sect. 2.1) values are also the lowest for the westerly winds (280°) and the highest for the south-south-easterly winds (160°). The absolute maximum in the mean $NO_2$ VCD values is also observed for the 160° wind direction at 200–300 m, although this maximum is not as well-pronounced as in the case of $SO_2$. This may suggest that the $NO_2$ and $SO_2$ are coming from the same source, although there are other $NO_2$ sources in the area. Also, unlike $SO_2$, $NO_2$ VCD is also high from the 40° wind direction, i.e., from the north-eastern mining area (see Fig. 2b). Compared to the Pandora $NO_2$ observations, the in situ $NO_2$ data show more different results for its vertical sensitivity. Although the maximum $NO_2$ amount is still from 160°, in situ data is more sensitive to the wind layer heights in 50–100 m (Fig 4b, 160° result). A clear decreased sensitivity can be seen above 100 m with minimum values around 200–300 m. The results indicate Pandora is more sensitive to transported $NO_2$ than in situ instruments, while in situ instrument is more directly linked to preferred wind conditions (i.e., the near-surface wind must be from 160°). For the nearby source, i.e., 40°, both Pandora and in situ observations show similar straight vertical structures (see blue lines in Fig. 4). For the background observations (i.e., 280° and 340°, low $NO_2$ conditions), both Pandora and in situ observations show similar sensitivity for surface layer winds.

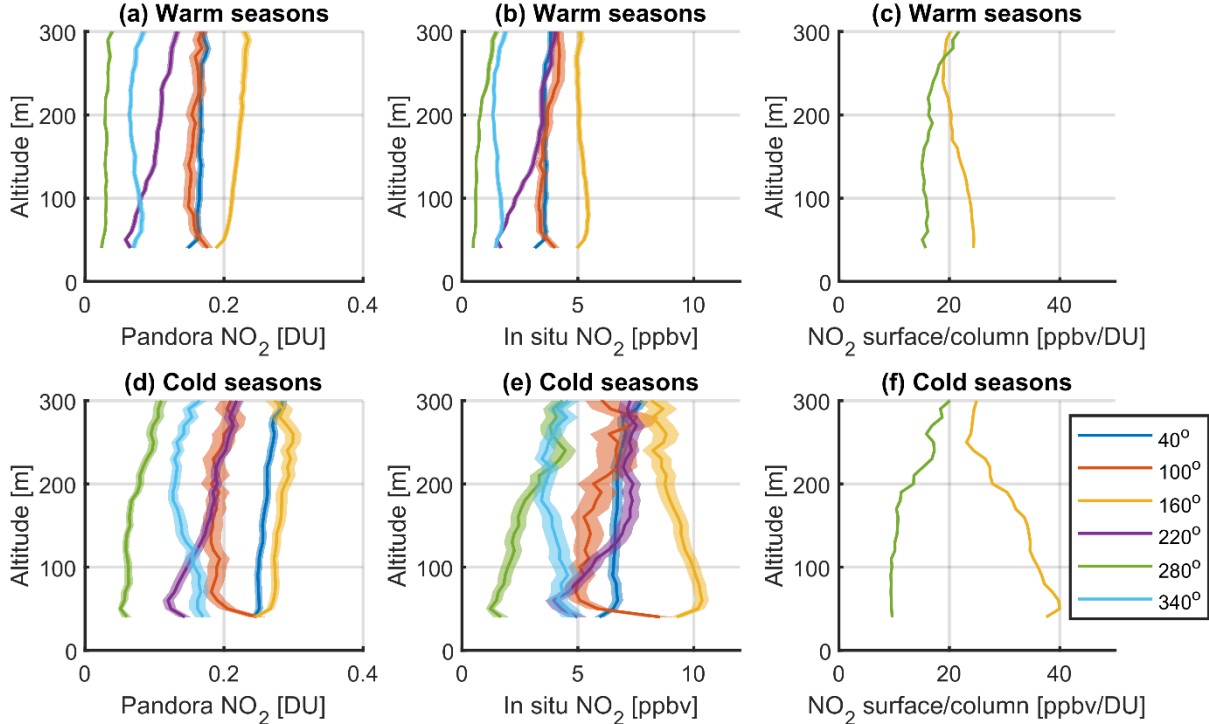

**Figure 4. The mean NO₂ column (a and d), surface (b and e) observations, and their median ratios (c and f) binned by wind directions at different altitudes at Oski-Otin. Wind layers information (altitude and direction) were measured by WindRASS, NO₂ column observations were from Pandora, surface measurements were from in situ instrument. Color lines show the mean values of NO₂ with 1-sigma envelopes. Data were categorized by warm (May to Oct.; top row) and cold (Nov. to Apr.; bottom row) seasons.**

An interesting feature that both in situ $SO_2$ and $NO_2$ analyses demonstrated is an enhanced sensitivity above 150 m in 220° wind directions (Figs. 2b and 4b, purple lines). As the 220° wind bin covers the tailing ponds area and a large portion of the mining area (observations within 190°–250° wind directions), it is likely the observed transported emissions were from these areas. Thus, such 60° wind bins might be too coarse to separate some sources (fine bins were used to reveal horizontal transport in Sect. 4, and no prominent emission sources were identified in 220° wind directions).

In general, by utilizing various ground-based remote sensing and in situ observations, such vertical sensitivity analysis can reveal the characteristic of air pollutants sources more efficiently than using in situ instrument only. When comparing the analysis results categorized by warm and cold seasons, the structures of vertical sensitivities show the changes due to different vertical dynamics. For example, in situ surface concentration data show more enhanced sensitivity for near-surface wind in cold seasons than in warm seasons (Figs. 4b and 4e). However, even in cold seasons, Pandora data still preserved its sensitivity to transported emissions (200–300 m winds, compare yellow lines in Fig. 4d and 4e), as it measures the integrated vertical columns. When comparing Figs. 4c and 4f, the surface to column ratio at 70 m changes from 39 ppbv/DU in cold seasons to





24 ppbv/DU in warm seasons. The difference between polluted and clean air is also much larger in cold seasons, indicating a stronger inhomogeneity of tropospheric $NO_2$.

### 3.3 Sensitivity of aerosol observations to winds at different levels

The sensitivity analysis of aerosol data to the wind speed and direction is shown in Fig. 5. In contrast to $SO_2$ and $NO_2$ data,
observations of aerosol data show greater differences between AOD and in situ $PM_{2.5}$ measurements, when they are binned by the wind direction. Similar to the Pandora instruments, the sunphotometer samples the aerosol through a slant light path (i.e., direct-sun viewing geometry, sample the atmosphere constituents between the instrument and the sun) and reports the vertically integrated column property. However, establishing the link between the AOD and $PM_{2.5}$ is more difficult (Sioris et al., 2017), compared to the work for trace gases observations since the AOD additionally depends on aerosol composition, details of the
size distribution, and even aerosol shape.

Similar to the $SO_2$ and $NO_2$ data analysis, the in situ $PM_{2.5}$ data show maximum sensitivity to the winds from 160°, indicating that most fine particles were from the direction of upgraders. However, the spread between the aerosol values from different directions is noticeably less than that for $SO_2$ and $NO_2$. But the vertical structure of $PM_{2.5}$ dependence on the 160° winds as a function of the wind height (Fig. 5b, orange line) is nearly the same from the surface to 300 m layers. Thus, the
$PM_{2.5}$ concentration under 160° wind direction was not sensitive to a particular layer of the wind. Comparing to the profiles of in situ $SO_2$ (elevated at 200–300 m; emissions from high stacks) and in situ $NO_2$ (two peaks, one near-surface and one elevated at 200–300 m; emissions from both near-surface mining fleet and high stacks), the sources of $PM_{2.5}$ from this direction could not easily be distinguished. Comparing Figs. 5a and b, AOD data show higher sensitivity to winds from 100°, while $PM_{2.5}$ data show higher sensitivity to winds from 160°. These inconsistent results might be because the wind bins were selected based on
our knowledge of $SO_2$ and $NO_2$ sources, i.e., 100° and/or 160° wind bins might be mixed with several different sources of aerosols in this region. The AOD and in situ observations are in better agreement for other wind directions, i.e. they both show clean air (low aerosol) was from 280°.

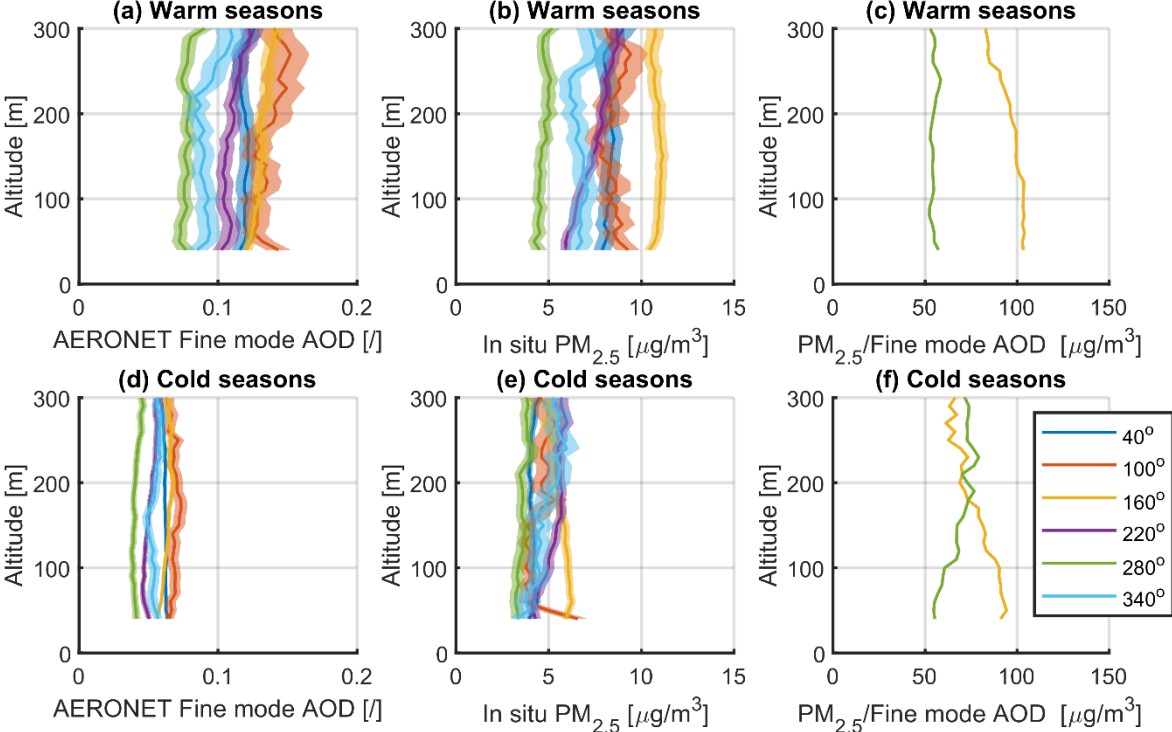

**Figure 5. The mean aerosol column (a and d), surface (b and e) observations, and their median ratios (c and f) binned by wind directions at different altitudes at Oski-Otin. Wind layers information (altitude and direction) were measured by WindRASS, aerosol column observations (AOD) were from sunphotometer, surface measurements (PM$_{2.5}$) were from in situ instrument. Color lines show the mean values of aerosol with 1-sigma envelopes. Data were categorized by warm (May to Oct.; top row) and cold (Nov. to Apr.; bottom row) seasons.**

In contrast to NO$_2$ results, which have increased surface concentration values in cold seasons, aerosol results reflect opposite patterns. AODs are smaller in cold seasons from all wind directions compared to warm seasons (see Figs. 5a and 5d). In situ PM$_{2.5}$ values also show similar decrease from warm to cold seasons, especially from the pollution transport direction (i.e., 160°; see Figs. 5b and 5e). More importantly, no clear vertical structure changes were found for aerosol observations in different seasons. One reason could be that the vertical structure of aerosols does not change for different seasons in this area, or the instruments are not sensitive enough to reveal such changes (i.e., the precision of aerosol measurements and wind data). Another reason is that there are some high background aerosols that do not depend on the wind direction. In short, Fig. 5 reveals that the biases between AOD and PM$_{2.5}$ data are different in different seasons. The surface to column ratio of aerosol data (see Figs. 5c and f) show higher than background values from 160° directions for both cold and warm seasons. Such ratio is lower in cold seasons (91 µg/m$^3$ vs. 103 µg/m$^3$ at 70 m in cold and warm seasons, respectively) and intersect with background





conditions (green line) at 180 m (Fig. 5f), indicating the aerosols layers are less sensitive to winds above this height in cold seasons. More discussion and interpretation of such ratio values are provided in Sect. 5.2.

## 4. Horizontal transport differences

### 4.1 SO₂ transport observations

Information about the wind speed in addition to wind direction could further help to characterize the pollution sources. In this section, the $SO_2$ observations from Pandora and in situ instrument are binned by the wind direction and speed data at a specific wind layer from WindRASS (i.e., $SO_2$ data binned by wind directions using 20° bins and by wind speed using 1 m s$^{-1}$ bins, at a certain height). The $SO_2$ data are plotted using polar coordinates where the radius refers to the wind speed and angle refers to the wind direction. The color of the shading areas represents the VCD or surface concentration of $SO_2$. For example,

Fig. 6a shows the Pandora $SO_2$ vertical column data displayed in such polar coordinates. The wind information for this plot was from WindRASS data at a height of 250 m, as indicated on the plot. The wind layer was chosen as both the remote sensing and in situ observations show maximum sensitivity of the $SO_2$ plume at this layer (see Fig. 3). Figure 6a shows the pollutants ($SO_2$ VCD) were transported from 130° to 190° wind directions and reached maximum values at a wind speed of about 7 m s$^{-1}$. Low wind speed conditions (e.g., < 4 m s$^{-1}$) within this direction range will have lower observed signals compared to higher

wind speeds. In contrast, Fig. 6b shows the in situ observations are sensitive to similar wind directions but elevated $SO_2$ surface concentrations are less sensitive to the wind speed than VCDs.

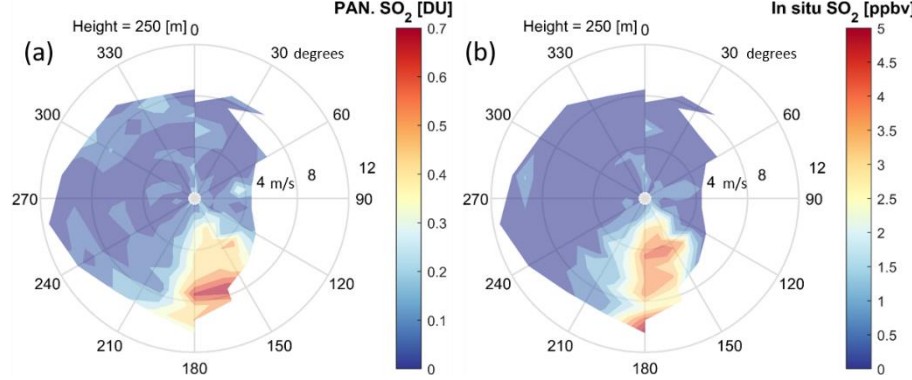

**Figure 6. SO₂ horizontal transport pattern resolved by wind speed and directions. SO₂ observations from (a) Pandora (SO₂ VCD in DU) and (b) the in situ instrument (SO₂ surface concentration in ppbv) at Oski-Otin. Wind information from WindRASS**
**data at 250 m layers where both Pandora and in situ instrument show maximum sensitivity to wind from the pollution sources (from 160°). The polar plot bins the SO₂ data (color-coded shading areas) by wind directions (angle values in degrees) and wind speed (radius values in m s⁻¹).**

Both Pandora and in situ instruments show the same behaviour with wind speed: high $SO_2$ for relative higher wind speed (4–8 m s$^{-1}$) and low for low wind speed (<1 m s$^{-1}$). This is simply because there are no local $SO_2$ sources in Fort McKay.



The higher the wind speed, the less time the pollutants need to travel from the source to the measurement site. Since the upgraders are about 20 km away from the observation site, fast winds would be corresponding to a transit time of about 1 hour; while slow winds would corresponding to about 6 hours. As $SO_2$ has a relatively short lifetimes in lower troposphere (of a few hours), the less time it takes to travel, the higher are the observed VCD or surface concentrations. If the wind speed is even

higher (>8 m s$^{-1}$), the $SO_2$ signal is low again since the pollutant is spread over a larger area, while its total mass, determined by the emission rate and the lifetime, remains constant (e.g., see $SO_2$ VCD plume model from Fioletov et al., 2015).

## 4.2 NO₂ transport observations

For $NO_2$, the data were analyzed at two different wind layer heights, i.e., wind data from the 250 m layer (same as Fig. 6 which has high sensitivity to the wind direction for $SO_2$ data and remote sensing $NO_2$ data) and 60 m layer (high sensitivity for in situ

$NO_2$ data, as shown in Fig. 4). In contrast to the single source direction for $SO_2$ (160°), two distinct $NO_2$ source directions were found at 40° and 160° (Fig. 7a and 7c). As shown in Fig. 2, the $NO_2$ source from 40° is related to the mining area in the North West direction of the observation site ($NO_2$ emission mainly from mining fleet). The 160° directions have $NO_2$ emissions from both stacks and mining fleet (surface emissions >50%). The in situ instrument revealed the same source directions, but with different sensitivities to wind speed and wind layers. Fig. 7b shows that the 250 m layer wind is not good enough to isolate

$NO_2$ sources, if using the in situ $NO_2$ observations. When using the wind layer at 60 m, Fig. 7d shows that the in situ instrument has increased sensitivity to surface $NO_2$ at low wind speed conditions (i.e., < 4 m s$^{-1}$). The results confirmed that remote sensing and in situ instruments have different sensitivity to different layers of the pollutants, thus when comparing the results of their observations, extra caution should be given to account for such differences. Note that the directions of the winds with the highest $NO_2$ values are somewhat different between the two layers that may be related to vertical wind shear.

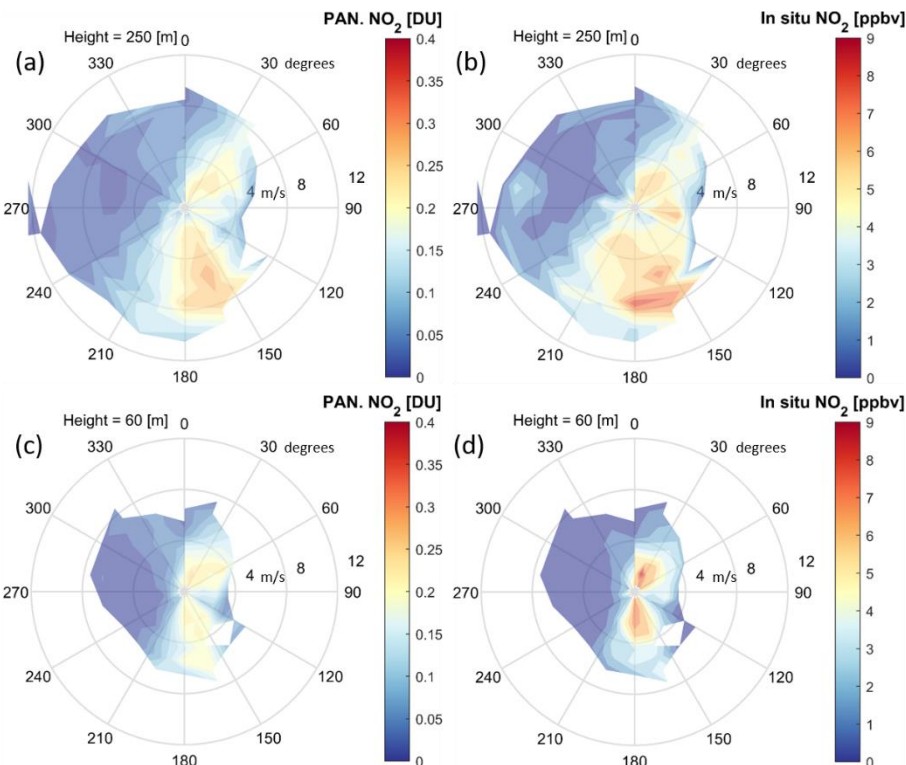

**Figure 7. NO₂ horizontal transport pattern resolved by wind speed and directions. NO₂ observations from (a, c) Pandora (NO₂ VCD in DU) and (b, d) in situ instruments (NO₂ surface concentration in ppbv) at Oski-Otin. Wind information from WindRASS data at 250 m layer (a and b) and 60 m layer (c and d). The polar plot bins the NO₂ data (color-coded shading areas) by wind directions (angle values in degrees) and wind speed (radius values in m s⁻¹).**

### 4.3 Aerosol transport observations

As illustrated in Fig. 5 and discussed in Sect. 3.3, there are larger differences between AOD and in situ measurements in determining the aerosol pollution directions. It is worth noting that there were no clear vertical structures that can be used to identify the optimal wind layers to separate and isolate potential aerosol sources. As a result, various wind layers and wind bins were examined. Not surprisingly, we did not find an optimal wind layer or wind bins that shows agreement between remote sensing and in situ measurements (see Fig. 8; which shows only warm seasons, when the aerosol loads are high). However, some general patterns can still be observed. In situ data show a clear aerosol source at 160° and other sources from the North and North East (see Fig. 8b and 8d; 250 m and 60 m wind layers, respectively), but with slightly different patterns in terms of wind speed. The source in the North-East directions can be detected in high wind conditions (e.g., about 7 m s⁻¹), whereas the source in South East directions can be detected in almost all wind conditions. This could indicate different distances of the source of PM₂.₅ and the observation site, i.e., the South East source is closer. It is also worth noting that remote sensing instruments revealed those two aerosol sources, e.g., Fig 8a. However, the remote sensing observations demonstrate



elevated AODs only for high wind conditions. A better explanation of such difference between sunphotometer's fine mode AOD and in situ instrument's PM$_{2.5}$ observations is still needed.

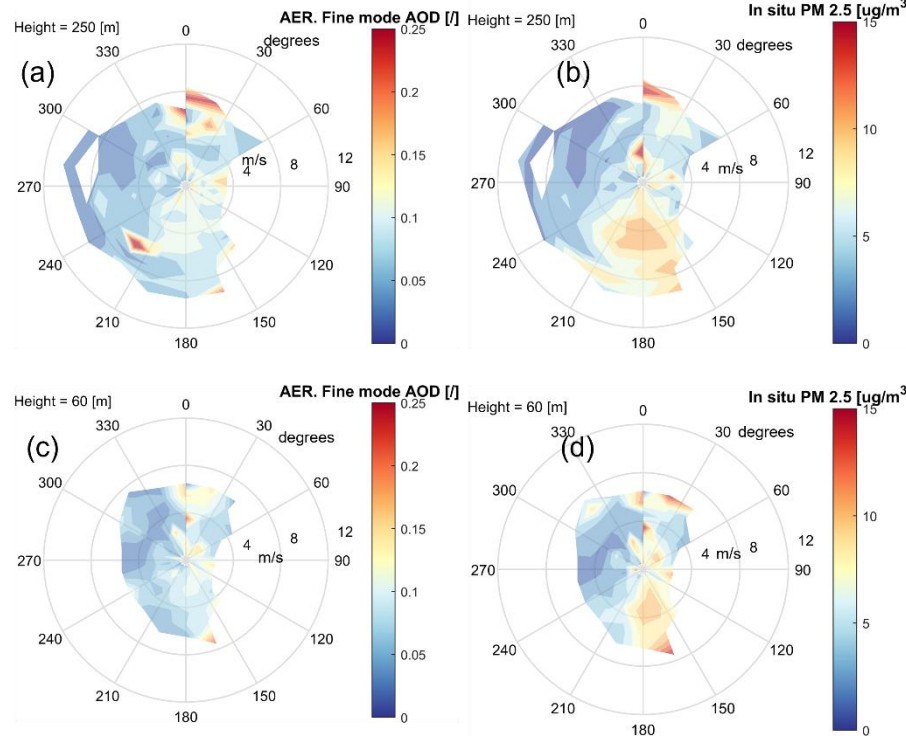

**Figure 8. Aerosol horizontal transport pattern resolved by wind speed and directions. Aerosol observations from (a and c) sunphotometer (fine mode aerosol optical depth, unitless) and (b and d) the in situ instrument (PM$_{2.5}$ surface concentration in µg m$^{-3}$) at Oski-Otin. Wind information from WindRASS data at 250 m layer (a and b) and 60 m layer (c and d). The polar plot bins the aerosol data (color-coded shading areas) by wind directions (angle values in degrees) and wind speed (radius values in m s$^{-1}$).**

## 5. Reanalysis meteorological data

Sections 3 and 4 demonstrated that vertical wind profiling information could play important roles in air quality monitoring and pollution transportation studies. However, such instruments (e.g., WindRASS) only provide limited coverage spatially (not many Pandoras or other air quality monitoring instruments are co-located with wind profiling instruments) and vertically (as discussed in Sect. 3, this WindRASS instrument only had good vertical coverage from 40 m to about 300 m above ground level (a.g.l.)). On the other hand, reanalysis meteorological data (e.g., ERA-5) have been used extensively in many research applications (e.g., Hersbach et al., 2020) and reanalysis wind data have quality acceptable for many applications and the advantage of good temporal coverage and global spatial coverage. For example, in our previous work, ERA-5 winds from 1000 to 900 hPa were averaged to provide wind direction and speed information to facilitate satellite validation and regional



air quality monitoring (Zhao et al., 2020, 2022). However, these wind layers may not always be optimal for different conditions (i.e., different species and/or emission sources heights as illustrated in previous sections).

In this part, we present a similar analysis as Sects. 3 and 4, but instead of WindRASS observations, ERA-5 model level wind data are utilized (from ERA-5 model levels 100 to 137; note level 137 is the bottom/ground level). This part of the study is to: 1) verify if ERA-5 winds can distinguish the pollution vertical and horizontal transport patterns as well as WindRASS, and 2) provide knowledge on the optimal ERA-5 wind layers that can be used for satellite-based pollutant emission estimations.

## 5.1 Wind analysis with ERA-5 data

Figure 9 shows the vertically resolved sensitivity analysis for $SO_2$ (first row), $NO_2$ (second row), and aerosol (third row) observations (similar to the results in Sect. 3, but utilizing ERA-5 wind data). To make the analysis comparable to the WindRASS data, the hourly ERA-5 model level wind data were interpolated to 100 m vertical resolution. The first two columns of Fig. 9 show the ERA-5 model results that represent the 0 m to 2000 m range (a.g.l), with only wind directions of 160° and 280°.

Figures 9a and b show that the $SO_2$ VCD and surface concentration observations show similar sensitivity to certain wind layers when ERA-5 winds are used instead of WindRASS data. For example, in warm seasons (see solid lines), both VCD and surface concentration of $SO_2$ show higher sensitivity to winds below 2 km; while in cold seasons, these observations of $SO_2$ show higher sensitivity to winds only below 0.5 km (see dashed lines). This seasonal difference was also found when using WindRASS data (see Sect. 3.1). Note that a direct comparison between ERA-5 winds and WindRASS observations results is challenging due to limited spatial (0.25° × 0.25°, approximately, 30 km × 15 km at Fort McKay) and vertical resolution of ERA-5 data (60–90 m below 500 m compared to 10 m for WindRASS).

In Figs. 9a and b, one major difference between ERA-5 and WindRASS results is the maximum sensitivity height. For $SO_2$ VCD observations, when using WindRASS, it has maximum sensitivity to winds from 250–300 m altitude (a.g.l; see Sect. 3.1). This could be due to the fact the WindRASS could not provide good sampling above this layer. When using ERA-5, $SO_2$ observations show a clear change in its maximum sensitivities for wind layers from 0–1400 m in warm seasons down to 0–500 m in cold seasons. Gordon et al. (2018) and Davis et al. (2020) show the retrieved $SO_2$ vertical profiles from MAX-DOAS and aircraft measurements with plume heights could reach 600 m to 1000 m altitude. Note that WindRASS only provides good samplings for wind layers at altitudes below 400–500 m.



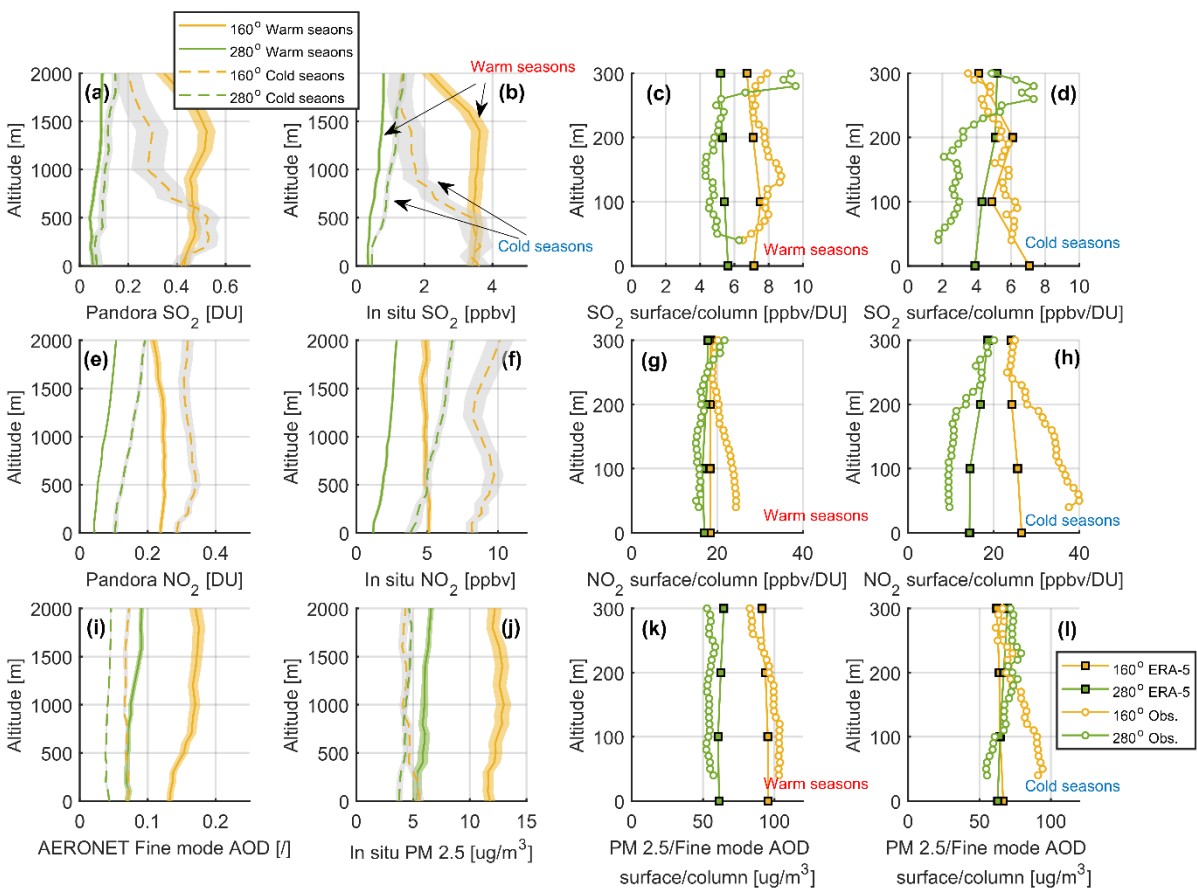

**Figure 9. The mean SO₂, NO₂, and aerosol as a function of the direction of winds (only 160° and 280°) at different altitudes from Pandora and in situ instruments at Oski-Otin. Other details of this figure are the same as Fig. 3, but the wind data is from ERA-5 reanalysis data (first two columns), not the WindRASS observations. Last two columns show the surface to column ratio for selected wind directions; solid lines with square symbols use wind data from ERA-5 reanalysis, lines with circles symbols use wind data from the WindRASS observations.**

For NO₂, Figs. 9e and f show that the largest NO₂ source is from 160° while the clean air is from 280° wind directions, consistent with findings in Sect. 3.2. For cold seasons, the NO₂ sensitivity height from 160° directions is consistent with the SO₂ results, indicating they both emitted from similar high stacks from these directions.

When using ERA-5 winds, the results for aerosol are also generally consistent with previous findings in Sect. 3.3. Figures 9i and j both confirm that there is a PM₂.₅ source from the 160° direction. Figures 9i and j show that this aerosol source can be better identified, if using winds from 500–1500 m layers. This feature is also partially reflected in Fig. 5a but not as clear as Fig. 9i due to the low data availability of WindRASS for higher altitudes. In general, Figs. 9i and j shows both AOD and PM₂.₅ data do not have strong changes in their vertical sensitivity to different layers of winds; i.e., we find no clear "plume"



heights changes for aerosol, in contrast to what we saw for $SO_2$ and $NO_2$ results. This result indicates that the aerosol is better vertically mixed than $SO_2$ and $NO_2$ emissions from high stacks. More details of horizontal transport differences when using ERA-5 winds are shown in Appendix B.

Regarding the surface to column ratios (0–300 m), the last two columns of Fig. 9 show a general agreement between
the results based on WindRASS (see circles symbols) and reanalysis (see square symbols) data, although the altitudinal dependence has some differences. For $SO_2$, the higher surface to column ratios from 160° directions in cold seasons (Fig. 9d) is generally preserved when using ERA-5 winds. For example, in cold seasons, the highest surface to column ratios from 160° directions is 7 ppbv DU$^{-1}$ (at 0 m altitude) and 6 ppbv DU$^{-1}$ (at 90 m altitude), for using ERA-5 and WindRASS measurements respectively. Overall, for $SO_2$, ERA-5 results generally preserved the features of ratio changes. For $NO_2$, the higher pollutant
to clean ratio differences are found in cold seasons (Figs. 9g-h), which is similar to the results when using WindRASS measurements. However, the larger discrepancy is found near the surface, where WindRASS results show higher ratios from 160° directions. These differences are mainly due to the coarse spatial and vertical resolution of ERA-5 data (i.e., only four data points for this 0–300 m range) and sampling issues with WindRASS measurements. The results for aerosol are similar to $NO_2$, and has better agreement in warm seasons.

**5.2 Boundary layer height effect on the column to surface ratio**

As shown in Sects. 3 and 5.1, for tropospheric pollutants, the surface to column concentration ratio can reveal some information about the vertical distribution. One important finding is that such ratios change with wind directions and also has seasonal patterns (e.g., see Fig. 9). The boundary layer height data from the lidar observations could provide critical information to help further interpret the difference between the vertical column and surface observations.
To better illustrate the effect of low BLH conditions on the $SO_2$ plume's vertical distribution, in this section, we plotted the column to surface ratio. For example, Fig. 10 shows the column to surface ratio binned by boundary layer height in warm seasons. Here the column values ($SO_2$ and $NO_2$ VCD, in the unit of DU; AOD is unitless) are from remote sensing observations, while $SO_2$, $NO_2$, and $PM_{2.5}$ surface concentration values ($SO_2$ and $NO_2$ in the unit of ppbv; $PM_{2.5}$ in unit of µg m$^{-3}$) are from in situ measurements and the BLH values are from lidar observations. Figures 10a, d, and g show the boxplot of
all observations; Figs. 10b, e, and h show the results representing observations with winds from the second-largest $NO_2$ sources (i.e., winds from 40° ± 30°; facilities of Syncrude, Shell Canada Limited, Canadian Natural Resources Limited, and Imperial Oil Resources); Fig. 10c, f, and i represent winds from the largest $SO_2$ and $NO_2$ sources (i.e., winds from 160° ± 30°; facilities of Suncor and Syncrude). The results clearly show that, for the $SO_2$ plume from upgraders (see Fig. 10c), the $SO_2$ column to surface ratio is negatively correlated to BLH. In other words, in low BLH conditions, surface in situ and remote sensing
observations will likely show more differences for such elevated plumes.

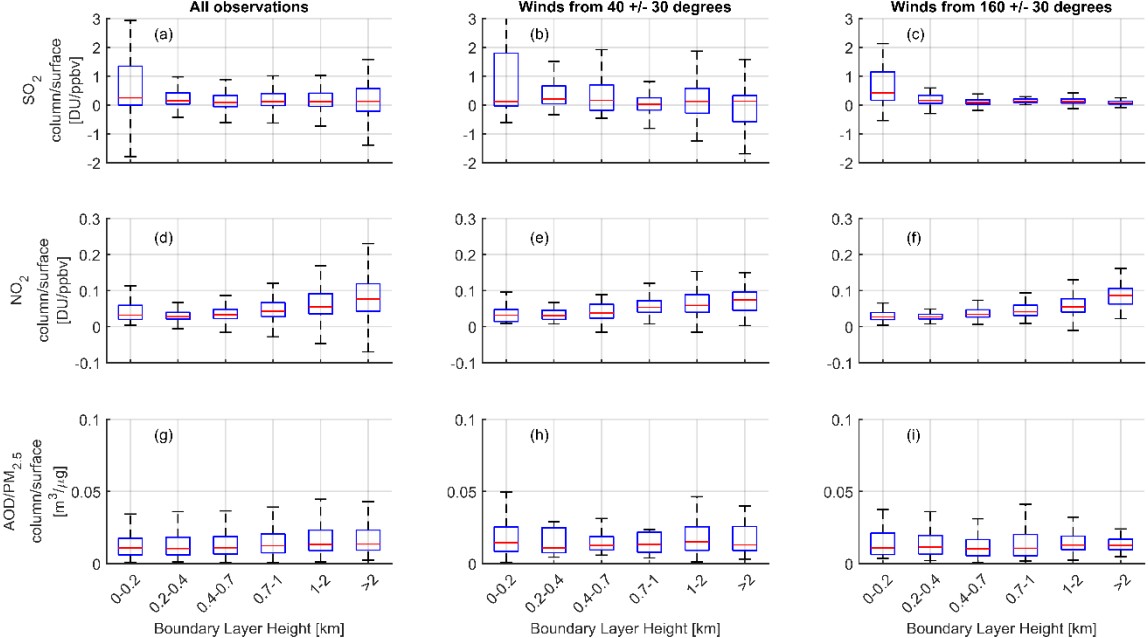

**Figure 10. Boxplot of SO₂, NO₂, and AOD/PM₂.₅ column to surface ratio binned by boundary layer height in warm seasons. Panel (a), (d), and (g) show all observations; panels (b), (e), and (h) show observations with winds from 40° ± 30° directions; panels (c), (f), and (i) show observations with winds from 40° ± 30° directions. For SO₂ and AOD, the wind data is from WindRASS observations at 250 m altitudes. For NO₂, the wind data is from WindRASS observations at 60 m altitudes. The boundary layer height is measured with lidar. The SO₂ and NO₂ column observations are from Pandora, AOD observations are from sunphometer, and all surface concentrations are from in situ instrument. On each box, the red central mark indicates the median, and the bottom and top edges of the blue box indicate the 25ᵗʰ and 75ᵗʰ percentiles, respectively. The black whiskers extend to the most extreme data points not considered outliers.**

In contrast, a similar analysis for NO₂ observations shows different features (see Figs. 10d-f). This time, as NO₂ has two major source directions (40° and 160° directions), Figs. 10e and f show clear and similar column to surface ratio patterns, i.e., this ratio is positively correlated with BLH. This result indicates that the NO₂ plumes have different vertical distributions than SO₂ plumes (note that, different than SO₂ emissions from high stacks, the truck fleet in the mining area is one of the main NOₓ emission sources).





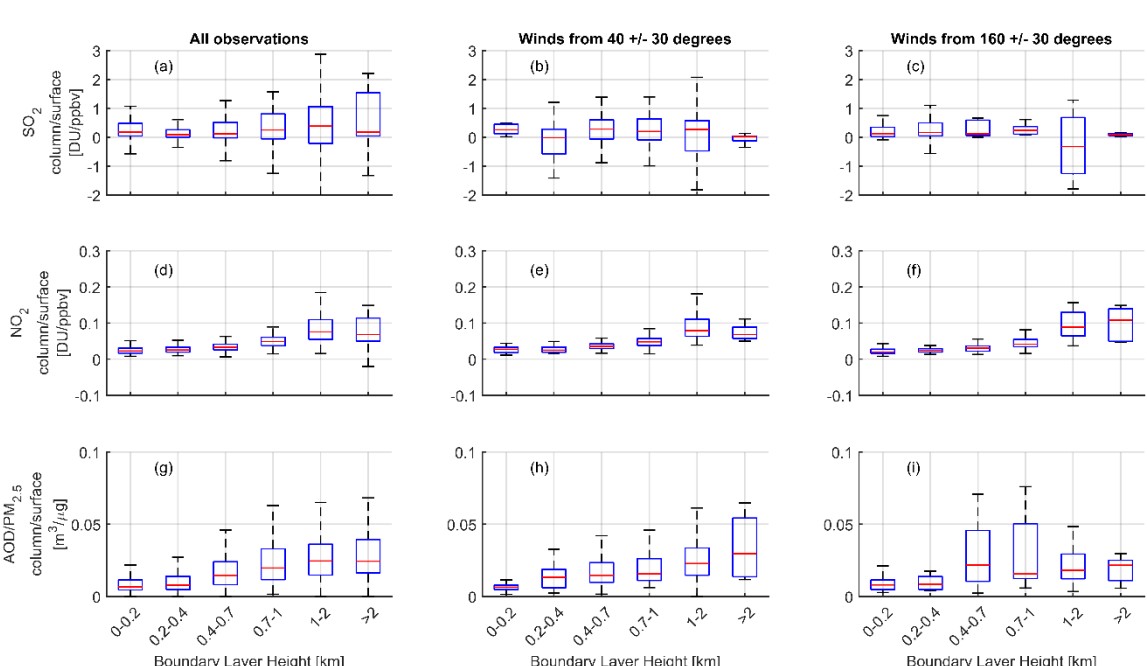

**Figure 11. Boxplot of SO₂, NO₂, and AOD/PM₂.₅ column to surface ratio binned by boundary layer height in cold seasons. Panel (a), (d), and (g) show all observations; panels (b), (e), and (h) show observations with winds from 40° ± 30° directions; panels (c), (f), and (i) show observations with winds from 40° ± 30° directions. For SO₂ and AOD, the wind data is from WindRASS observations at 250 m altitudes. For NO₂, the wind data is from WindRASS observations at 60 m altitudes. The boundary layer height is measured with lidar. The SO₂ and NO₂ column observations are from Pandora, AOD observations are from sunphotometer, and all surface concentrations are from in situ instrument.**

When performing the same analysis for AOD (from sunphotometer) and PM₂.₅ (from in situ measurements) observations, the results are not clear in warm seasons (see Figs. 10g-i), similar as in Sects. 3.3 and 4.3. I.e., no simple column-to-surface ratio correlations were found from 40° and 160° directions. Such results indicate the aerosol from two pollution sources (40° and 160° directions) are well mixed, consistent with findings in Sect. 3.3 (i.e., no vertical sensitivity changes when using different wind layers). However, in cold seasons, there is a positive correlation between the AOD/PM₂.₅ ratio to BLH data from 40° direction (see Figs. 11h). Thus, in cold seasons, from this direction, the aerosol loads are closer to the surface than vertically mixed.

However, as discussed, WindRASS or lidar observations are not typically available for Pandora or sunphotometer sites. As a result, such pollutants' vertical distribution information is hidden within Pandora and sunphotmeter observed data. We tested by replacing WindRASS and lidar observations with ERA-5 modelled results (wind profiles and BLH), and repeated the analysis shown above. The results show ERA-5 data could also assist the work in terms of separated pollutants' signal from different sources (see Figs. B2 and B3).





## 6. Integration period differences

In contrast to in situ instruments that can obtain 24 hr continuous observations, passive remote sensing instruments (i.e., use the sun as a light source) can only deliver data when direct solar light can reach the instruments. Figure 12 shows the coincident observations from remote sensing (blue solid lines with error bars) and in situ (red solid lines with error bars) observations that have been averaged by the hour (local standard time). Both $SO_2$ and $NO_2$ data show good agreement in variation patterns except early morning. The prominent discrepancy in the morning could be related to larger changes in the mixing layer height and related vertical mixing conditions. For example, in cold seasons, with increased mixing layer height and vertical dynamics from the morning to noon, in situ $SO_2$ shows a strong increase from 0.40 ppbv at 6 am to 1.18 ppbv at 10 am (see Fig. 12). In contrast, the Pandora observations show an opposite decreasing trend; the VCD of $SO_2$ decreased from 0.32 DU at 6 am to 0.01 DU at 10 am. As discussed before, Pandora can sample the entire vertical column of $SO_2$ molecules, thus the increase of surface $SO_2$ concentration (from less than 0.5 ppbv at 6 LST to almost 1.5 ppbv at 10 SLT) in the morning (see Fig. 12b, in the warm seasons) is not due to increased emission ($SO_2$ VCD values are stable from 6 to 10 LST) but due to vertical mixing. Better vertical mixing conditions in the late morning to noon help the elevated $SO_2$ plume to reach the surface level. With a well-mixed boundary layer after 10 am, the variation pattern of surface and column values of $SO_2$ shared a similar pattern.

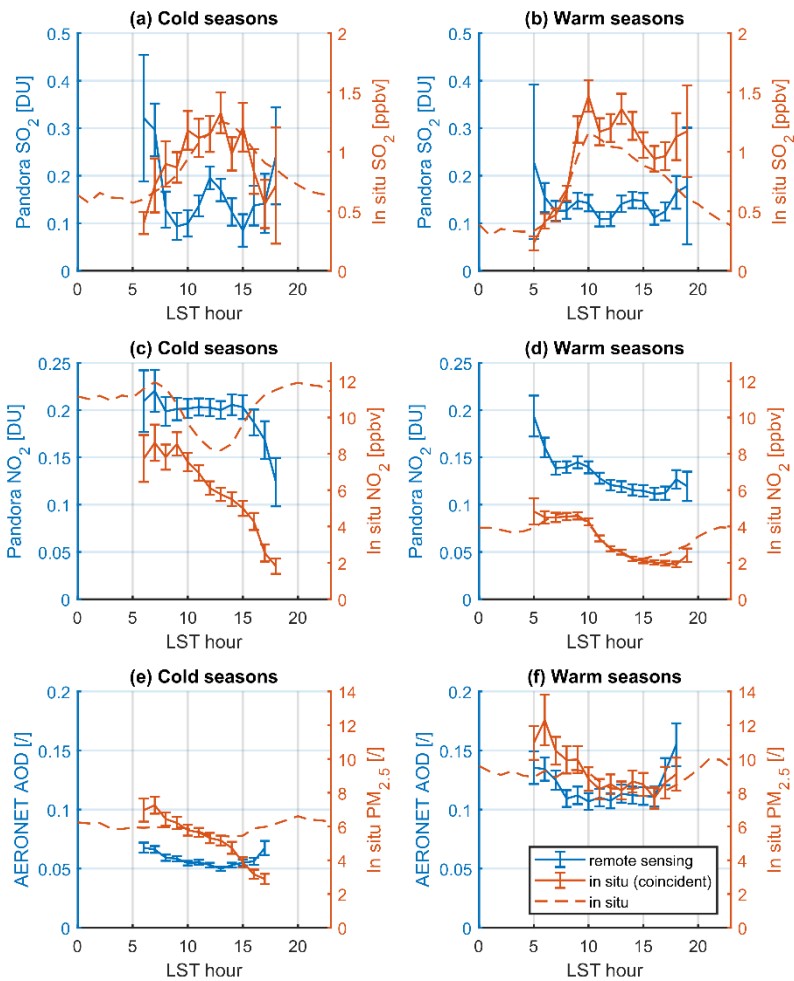

**Figure 12. Remote sensing and in situ observed SO₂, NO₂, and aerosol data (AOD and PM₂.₅) averaged by the hour of local standard time (LST). Remote sensing observation results are shown using the left axis (blue), in situ observation results are shown using the right axis (red). Solid red lines show in situ measurements that coincident with remote sensing observations; dashed red lines show all in situ measurements. Results from warm seasons are shown in panels (a), (c), and (e); results from cold seasons are shown in panels (b), (d), and (f).**

The conditions for $NO_2$ are similar to $SO_2$ but show stronger indications from photochemistry, i.e., generally decreased values of $NO_2$ from morning to evening. Moreover, as there are strong $NO_2$ sources near ground level, even the VCD are stable from 8 to 15 LST (e.g., see Figs. 12 b), the surface value is keep decreasing. For aerosol comparison, both AOD and $PM_{2.5}$ show clear U-shape changes over the day in warm seasons (see Fig. 12f). However, in the afternoons of cold





seasons, $PM_{2.5}$ shows a decreasing trend, while AOD has an increasing trend (see Fig. 12e). Thus, without detailed meteorological information (e.g., BLH and vertically resolved wind information), there is no surprise that we could not find an easy and clear correlation between remote sensing and in situ observations.

In addition to data sampled at the time of measurements by remote sensing instruments, results based on all in situ measurements are plotted in Fig. 12 as red dash lines. When comparing red dash lines with red solid lines, the results from Fig. 12 shows interesting and prominent features, i.e., when pairing in situ data with remote sensing data, sometimes, the diurnal patterns of in situ measurements are changed (e.g., see Fig. 12c, the different variations of the red solid and red dashed lines). The cause of such changes is due to the "sampling bias" from remote sensing measurements that are available under unobscured sun conditions only; in situ data has no major gaps. Although the local emissions conditions might not be different in sunny or rainy days, the difference in meteorological conditions can affect the pollutants monitoring results. For example, Fig. 13 shows the ERA-5 boundary layer height data binned by hours of local standard time. The BLH data that coincident with in situ $SO_2$ measurements show clear seasonal difference (see blue lines in Figs. 13a and b); in cold seasons, the highest BLH data only reach 750 m, while in warm seasons it can reach 1500 m. The diurnal variations of BLH data show bell curves as in situ instruments performs sampling 24 hr per day in all weather conditions. However, the BLH data that coincident with remote sensing $SO_2$ measurements (see red lines in Figs. 13a and b) show more skewed shapes, i.e., higher BLH values in afternoons. Especially, in the wintertime (Fig. 13a), BLH data that coincident with remote sensing observations shows a much stronger increasing from 10 to 16 LTS than its counterpart that coincident with in situ measurements. Similar features can be seen for $NO_2$ and aerosol data. Thus, it is clear that remote sensing observations are biased to high BLH conditions, especially in afternoons. These results also explain why, when coincident with remote sensing observations, in situ diurnal patterns changed (see Fig. 12). For example, in cold seasons at 17 LST, by pairing $PM_{2.5}$ data with AOD data, the corresponding BLH value increased from 500 m to 1500 m (see Figs. 13e) and thus the $PM_{2.5}$ values changed from 6.0 µg m$^{-3}$ to 2.9 µg m$^{-3}$ (see Fig. 12e). Such variation in diurnal patterns is not obvious in warm seasons, due to less BLH changes (due to coincident selection with remote sensing instrument) than cold seasons. For instance, by selecting coincident measurements, measured $NO_2$ surface values dropped from 11.3 pptv to 2.5 pptv in cold seasons at 17 LST (Fig. 12c); while the corresponding values only dropped from 2.5 pptv to 2.0 pptv, in warm seasons (Fig. 12d). In other words, the sampling biases between remote sensing and in situ data are worse in cold seasons than warm seasons. When direct comparing coincident remote sensing and in situ observations, such weather condition biased sampling would not easily been revealed. However, this clear-sky bias could lead to significant differences in long-term trend analysis.

In short, similar to findings in Sects. 3 to 5, Figs. 12 and 13 show that depending on the height of the emission sources, boundary layer dynamics can play important roles in the sampling differences between remote sensing and in situ instruments. Modelling of boundary layer conditions is also known to be a major difficulty for air quality modelling work (e.g., Zhao et al., 2019). This indicates a great need for improved boundary layer height observations from other remote sensing instruments





(Szykman et al., 2019). This issue could be more prominent if the remote sensing air quality measurements are performed in the urban area, where rush hour air pollutants are often more serious on cold winter. All these bring great challenges to utilizing remote sensing air quality observations for application to "noise-height" air quality monitoring and more efforts are needed.

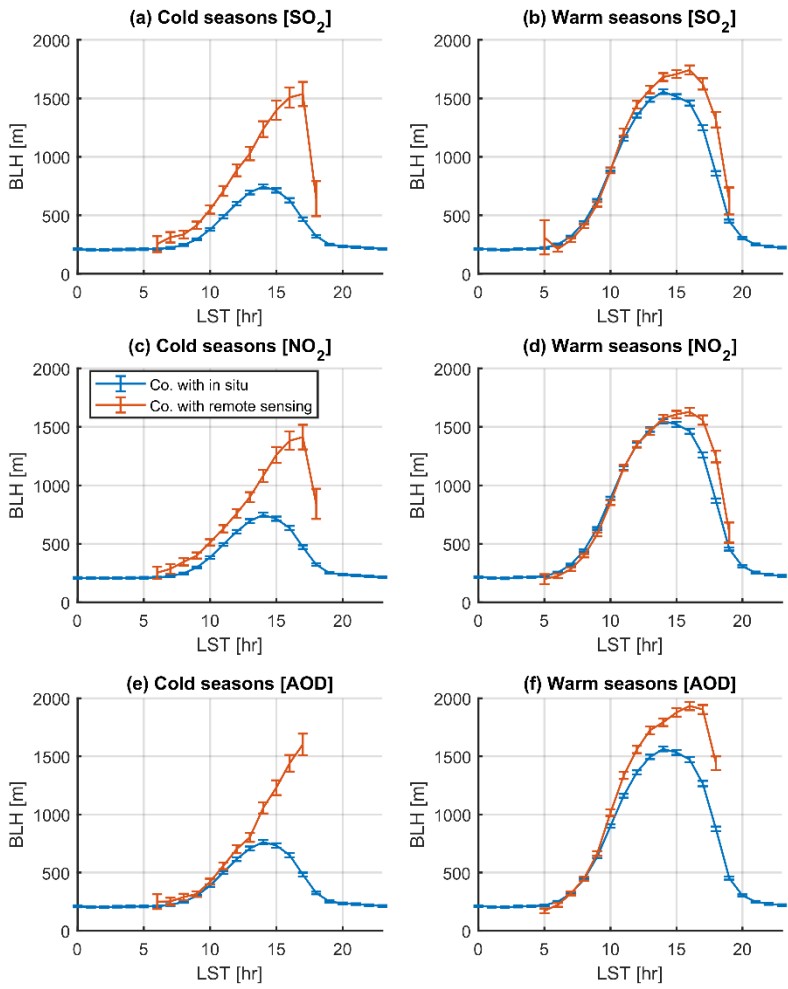

**Figure 13. Boundary layer height (BLH) data averaged by the hour of local standard time (LST). Blue lines show BLH mean values that based on all data; red lines show BLH data that coincident with remote sensing observations, i.e., when the Sun was not obscured. Results from cold seasons are shown in panels (a), (c), and (e); results from warm seasons are shown in panels (b), (d), and (f).**





## 7. Conclusion

This work analyzed the sampling differences (of $SO_2$, $NO_2$, and aerosol) between remote sensing and in situ instruments in the Canadian oil sands region, in terms of vertical, horizontal, and temporal dependency on meteorological conditions (wind speed, directions, altitude, and BLH). Results show that depending on the height of emission sources, remote sensing and in situ

instruments could sample different parts of pollutant plumes, due to their different vertical sensitivity ranges. For elevated pollutants (e.g., $SO_2$ emission via high stacks), both remote sensing and in situ instruments show strong dependence on wind information above stack height level (i.e., around 250 m for this case). The magnitude of the $SO_2$ VCD and surface concentrations reach their maximum for winds from $160° \pm 30°$ directions at 200–300 m altitudes (which is about 38% and 27% higher than the value for winds from the same directions but of near surface winds, respectively). In contrast, the $NO_2$

emissions from $160° \pm 30°$ directions are from both high stacks and the mining fleets. As a result, $NO_2$ VCD shows a more uniform sensitivity to winds from near surface to up to 300 m (peak value at 260–290 m altitude), while in situ measured $NO_2$ surface concentrations show a strong sensitivity to near-surface winds (peak value at 60–70 m altitude). In cold seasons, the $NO_2$ surface to column ratio from 160° direction changes from 39 ppbv/DU at 70 m to 28 ppbv/DU at 200 m. Such key differences indicate the challenges in applying remote sensing observations in "breathing-height" air quality applications. In

other words, although VCDs measured by remote sensing instruments represent integrated pollutants that are sensitive to both surface and upper levels, the observations could not easily be separated or linked to localized surface value. On the other hand, while elevated air pollutants cannot be monitored at the surface via in situ measurements, they can still be captured by remote sensing instruments. Thus, these VCD observations could be more useful in quantifying total emissions than surface observations for such emission sources. Also, it is worth noting that elevated air pollutants that do not affect the air quality

immediately as they are above ground level, could still pose a health risk further downwind as the pollutants will eventually reach the surface via vertical mixing. Thus, remote sensing VCD observations could represent air quality conditions for a much larger area than localized surface in situ measurements.

The horizontal transport sampling differences show that local sources have the largest impact on observations when the wind speed is low and the pollutants are not transported far from the source (as in the case of some $NO_2$ emissions). For

elevated sources (e.g., $SO_2$ emitted from high stacks), the moderate wind is substantial to bring the pollutant to the measurement site. Compared to in situ instruments, remote sensing observations are more sensitive to higher wind speed conditions, i.e., transported pollutants.

Comparing AOD and surface $PM_{2.5}$ measurements is the most challenging work, as their vertical and horizontal patterns are not very alike. Such larger differences are expected as these two measurements (AOD and $PM_{2.5}$) do not necessarily

represent the same matter, not only due to the different sampling area but also due to large seasonal cycle and high background. Comparisons are easier for e.g. $SO_2$, where remote sensing measured VCD (in DU) represents the $SO_2$ within the entire vertical column; in situ measured surface concentration (in ppbv) represents the portion of $SO_2$ within the air mass near the surface.



Although many studies exist that directly compare these two measurements (AOD and PM$_{2.5}$) and often reveal positive correlations, there are no simple methods exists to directly connect or convert them (e.g., only empirical methods or via modelling means). In general, due to the complex nature of PM, the conversion from PM$_{2.5}$ to AOD (or vice versa) is not straightforward. Here we show that linking these two measurements could be even more complicated, as they have more

sampling differences than observations of trace gases. On the positive side, both remote sensing and in situ observations show consistent uniform sensitivities to the wind speed and direction from near surface to 300 m altitude, indicating the aerosol loads in this region are more uniformly mixed than SO$_2$ and NO$_2$.

This analysis also shows that ERA-5 model level winds produce results similar to those for direct wind measurements by WindRASS at this location and, therefore, can be a good tool to support ground-based remote sensing research to identify

pollutant sources' directions and even their vertical structures. Using measured wind profiles and BLH, the work demonstrated that the column to surface ratio of pollutants could show positive or negative correlations with boundary layer heights, depending on the height of emission sources. Further results show replacing measured wind profiles and BLH by ERA-5 data could also preserve these features. Thus, the boundary layer height and wind profile data from ERA-5 also can be utilized to reveal pollutants' vertical distribution and mixing conditions, which can be used as critical information when converting

remote sensing column data to surface values. Current results also suggest that, for wind-rotation SO$_2$ or NO$_2$ emission models (e.g., Fioletov et al. (2016, 2022))  and wind-rotation satellite data validation method (Zhao et al., 2020), different layers of ERA-5 winds should be averaged, and some seasonal changes might be considered to improve the results.

This analysis of surface to column ratios also shows that the column values cannot be converted to surface by just one value of the ratio. Depending on the wind direction, the ratio for directions related to the pollution sources could be a

factor of two larger than these from "clean" directions.

The overall outcome of this work is to reveal and analyse the fundamental sampling differences between remote sensing and in situ observations via utilizing vertical wind observations and boundary layer conditions. Other aspects, such as sampling areas and clear-sky bias still need further investigation. In particular, as remote sensing instruments discussed here are sunlight instruments and their observations are clear-sky biased, these could lead to differences in long-term trend analysis

done with in situ and remote sensing instruments alone. These ground-based remote sensing air quality observations (e.g., Pandora and sunphotometer) cannot replace surface in situ monitoring, but can play a key role in linking satellite air quality observations and surface in situ measurements. Having the ability to link satellite measurements to surface concentrations will benefit air quality monitoring in critical environmental areas where surface in-situ measurements are sparse or not available, and thus will allow for improved characterization of the state of the environment.

*Data availability.*



Pandora data are available from the Pandonia network (http://pandonia.net/data/). Sunphotometer data are available from https://aeronet.gsfc.nasa.gov/cgi-bin/data_display_aod_v3?site=Fort_McKay&nachal=2&level=3&place_code=10. WindRASS and lidar data are available from http://collaboration.cmc.ec.gc.ca/cmc/arqp/. Any additional data may be obtained from Xiaoyi Zhao (xiaoyi.zhao@ec.gc.ca). WBEA in situ observations data is available at https://wbea.org/network-and-data/monitoring-stations/. ERA-5 data can be obtained from https://cds.climate.copernicus.eu/cdsapp#!/dataset/reanalysis-era5-pressure-levels?tab=overview.

*Author contributions.* XZ analyzed the data and prepared the manuscript, with significant conceptual input from VF and critical feedback from all co-authors. DG and CMcLinden provided and supported the analysis of ERA-5 model level wind data. JD, VF, XZ, and SCL operated and managed the Canadian Pandora network. IA, VF, and SCL operated and managed the AEROCAN. AC and MT operated the Pandonia Global Network (PGN) and provided critical technical support to the Canadian Pandora measurement program and subsequent data analysis. RSw managed the NASA Pandora project and supported Canadian Pandora measurement at the Oil Sands site. RSt provided WindRASS data products, and CMi provided in situ concentration and meteorological data. KS provided lidar data products.

*Acknowledgements.* The authors would like to thank the Wood Buffalo Environmental Association (WBEA) for the provision of their in situ data. Pandora, sunphotometer, and WindRASS measurements were carried out as part of the Oil Sands Monitoring (OSM) program by the Governments of Alberta and Canada. We thank Akira Ogyu and Reno Sit from ECCC, Daniel Santana Diaz from PGN, and Nader Abuhassan from NASA for their technical support of Pandora measurements. The PGN is a bilateral project supported with funding from NASA and ESA. This work was partially funded under the Oil Sands Monitoring (OSM) Program. It is independent of any position of the OSM Program.





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





**Appendix A.**

Figure A1 shows WindRASS' successful sampling rate and number of successful observations as a function of altitude. The number of successfully observations was nearly complete below 200m, but decreased quickly from 200 m to 400 m (i.e., from about 80% to about 20%). Less than 5% of observed vertical wind profiles reach altitudes above 600 m.

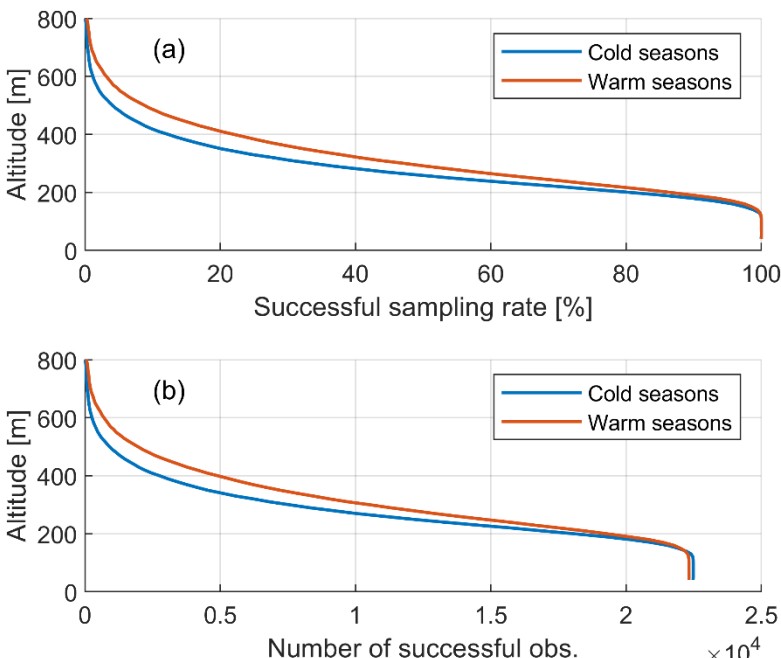

**Figure A1. WindRASS successful sampling rate (recovery rate) and number of observations as a function of altitude.**

**Appendix B.**

With the ERA-5 wind data, horizontal transport patterns were also analysed for $SO_2$, $NO_2$, and aerosol observations. The results are consistent with the results of Sect. 4. For example, Figs. B1a and b confirmed that Pandora $SO_2$ column data is more sensitive to high wind speed conditions than in situ surface $SO_2$ data. Similarly, Figs. B1c and d identified two $NO_2$ sources from expected directions. Also, Fig. B1d is consistent with Fig. 7d, both show the in situ $NO_2$ data is more sensitive in low wind conditions. It is worth noting that, for the aerosol data, the AOD sources can be better identified and separated (see the discussion in Sect. 5.1) for higher-level winds (1000 m). In general, it was found that the optimized ERA-5 wind layers for $NO_2$ pollutants transport analysis in this region range from the surface to 900 hPa year-round (corresponding to the bottom 15 layers in ERA-5 model level data). As the $SO_2$ emissions were from high stacks, in warm seasons, its optimized wind layers extended from the surface to 800 hPa (corresponding to the bottom 25 model layers); while in cold seasons, the optimized wind layers extended from the surface to only 900 hPa (the bottom 15 model layers).





With ERA-5 wind and BLH data, the boundary layer height effect shown in Sect. 5.2 has been reproduced in Figs. B2 and B3 for warm and cold seasons, respectively.

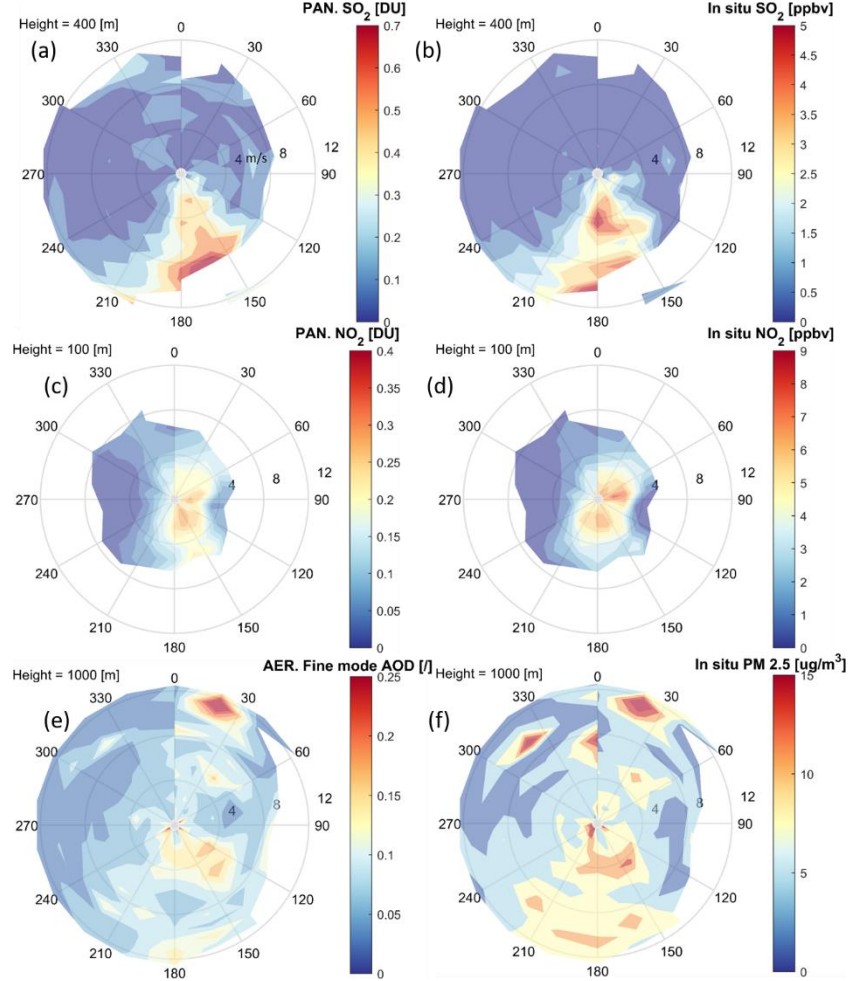

**Figure B1.** SO₂, NO₂, and aerosol horizontal transport patterns were resolved by wind speed and directions with using ERA-5 wind data. Panels a, c, and e show the SO₂, NO₂, and aerosol column observations from Pandora and sunphotometer (fine mode aerosol optical depth, unitless); panels b, d, and f show the in situ observations at Oski-Otin. Wind information from ERA-5 data at 400 m layers (a and b), 100 m layer (c and d), and 1000 m layer (e and f). The polar plot bins the observation data (color-coded shading areas) by wind directions (angle values in degrees) and wind speed (radius values in m s⁻¹).





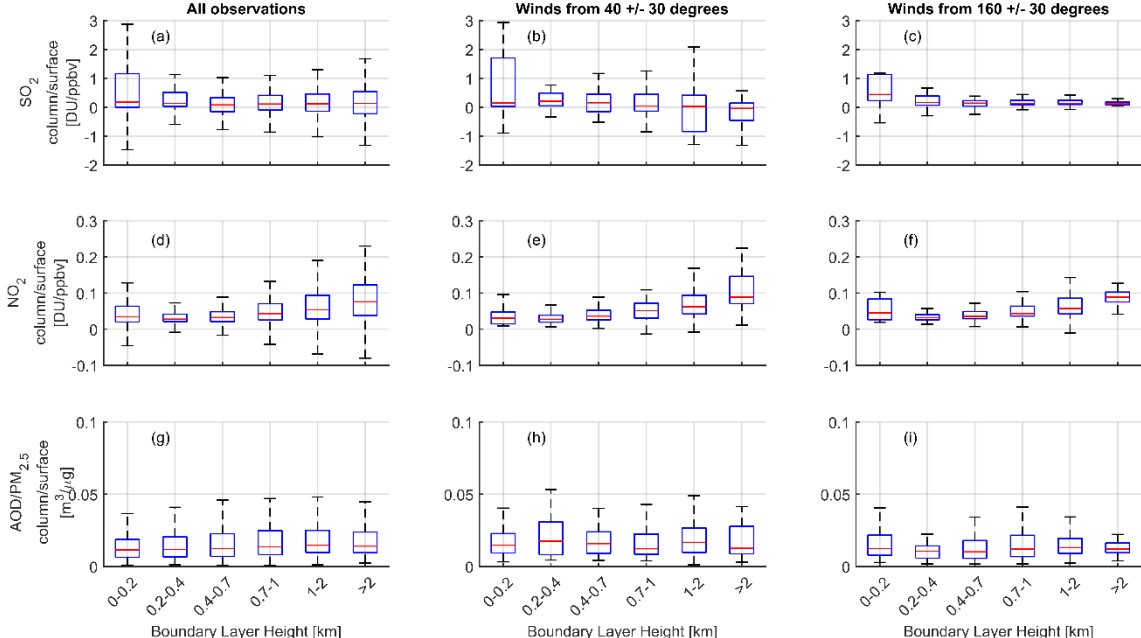

**Figure B2. Boxplot of $SO_2$, $NO_2$, and $AOD/PM_{2.5}$ column to surface ratio binned by boundary layer height in warm seasons. Panel (a), (d), and (g) show all observations; panels (b), (e), and (h) show observations with winds from $40° \pm 30°$ directions; panels (c), (f), and (i) show observations with winds from $40° \pm 30°$ directions. For $SO_2$ and AOD, the wind data is from ERA-5 at 300 m altitudes. For $NO_2$, the wind data is from ERA-5 at 100 m altitudes. The boundary layer height is from ERA-5. The $SO_2$ and $NO_2$ column observations are from Pandora, AOD observations are from sunphometer, and all surface concentrations are from in situ instrument.**





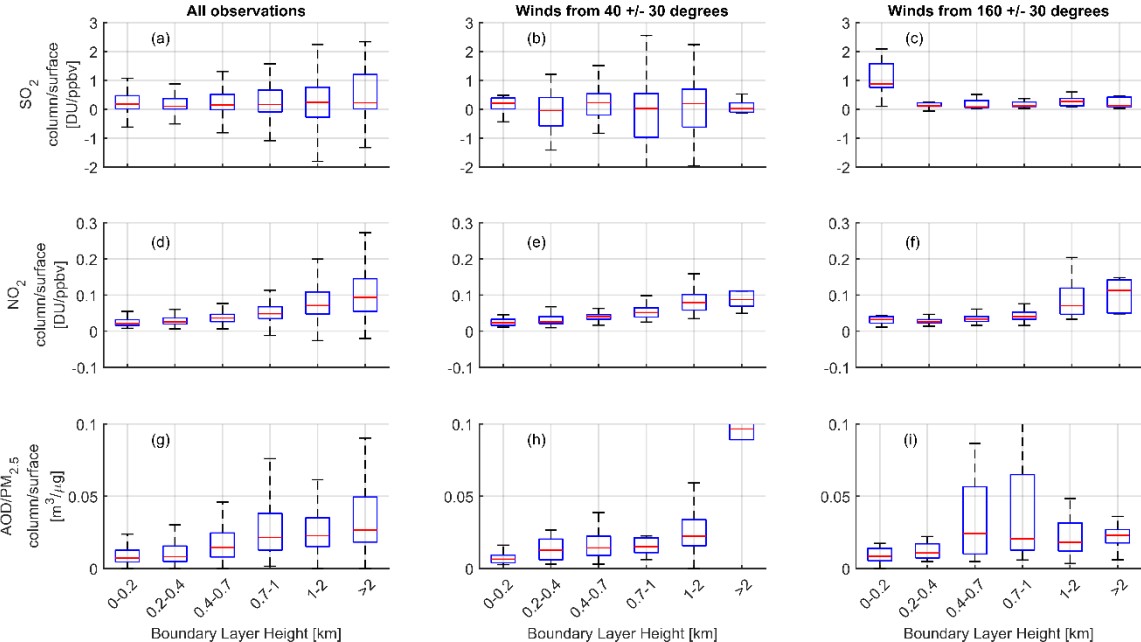

**Figure B3. Boxplot of SO₂, NO₂, and AOD/PM₂.₅ column to surface ratio binned by boundary layer height in cold seasons.**
**Panel (a), (d), and (g) show all observations; panels (b), (e), and (h) show observations with winds from 40° ± 30° directions; panels (c), (f), and (i) show observations with winds from 40° ± 30° directions. For SO₂ and AOD, the wind data is from ERA-5 at 300 m altitudes. For NO₂, the wind data is from ERA-5 at 100 m altitudes. The boundary layer height is from ERA-5. The SO₂ and NO₂ column observations are from Pandora, AOD observations are from sunphometer, and all surface concentrations are from in situ instrument.**