# Peer review of "The differences between remote sensing and in situ air pollutants measurements over the Canadian Oil Sands"

_Atmospheric Measurement Techniques, 2024_

## Referee Comment (RC2)

Reviewer comments for:

**The differences between remote sensing and in situ air pollutants measurements over the Canadian Oil Sands**

This review is for the above manuscript submitted for publication in Atmospheric Measurement Techniques. The manuscript focuses on two key intercomparisons for data collected at or near Canadian Oil Sands: 1. in situ instrumentation measuring surface-level concentrations of multiple criteria air pollutants ($SO_2$, $NO_2$, $PM_{2.5}$) and remote sensing instrumentation measuring total column concentrations of the same pollutants, and 2. Vertically resolved wind profiles measured at site and modelled wind profiles obtained from an international dataset (ERA-5). The authors conduct these comparisons primarily by linking altitude-based wind direction with concentration data. The authors find the presence of higher concentrations linked to certain wind directions irrespective of altitude for all three pollutants, and attribute these higher concentrations to key local sources in the region. Further, the authors also note the dominance of higher concentrations linked to certain altitudes (while fixing the wind direction), and present it as evidence of a strong influence of winds at that altitude on the measurement. The authors also repeat the analysis above with modelled wind data, and find similar results, suggesting that in absence of measured wind data, modelled data can be used. The authors also note the differing temporal coverage of data from in situ surface and remote sensing instrumentation, and quantify the biases in sampled and actual variations in BLH and pollutant concentrations. While the presentation of the altitude-concentration plots resolved by wind direction were very interesting, and the findings seem plausible too, I think that this manuscript lacks an underlying methological framework needed to conduct these comparisons. Additionally, the current choice of presentation is very difficult for the reader to digest, and requires substantial changes as well. I suggest the authors **revise and resubmit** this manuscript for the following major issues.

1. **Issues with motivation of the paper.**
The authors chose multiple lines to motivate this work. This includes "applying satellite measurements to surface air quality applications" such as "air pollution monitoring" and "linking satellite air quality observations and surface in situ measurements". However, it is unclear how the authors have addressed these specific knowledge gaps in the manuscript. I recommend that the authors list clear objectives in the introduction and address those

objectives step by step in different sections. This would also affect the abstract and the conclusions section, which would become more streamlined.

Additionally, the authors' literature review provides little motivation to compare in situ and ground based remote sensing data, with previous data showing bias <10% (Zhao et al., ACP 2019). If anything, the authors seem to have shown that in situ measurements combined with modelled ERA-5 data works really well, and that is a substantial finding. However, some of the other minutae can be removed from the conclusions then, with only key message kept in there.

**2. **Lack of uncertainty analysis**
In a paper like this with substantial use of modelled data, I think a detailed subsection on uncertainties of the different pollutants modelled or measured using different instruments is needed in the Methods section. I suggest that the authors also add a small table of the pollutants/meteorological variables measured/modelled, the instruments used to measure/model them, and their average concentrations, standard deviations, and uncertainties. Also, for atmospheric pollutant data, geometric mean and geometric standard deviation may be more representative (please check). For wind direction, median wind direction (and not mean) would be more representative. Similarly, all instances of mentioning specific concentration or DU values in the manuscript should be accompanied with associated uncertainty (e.g., on pg 10 line 4).

**3. **Comparison-based research without correlation coefficients?**
It was terribly inconvenient as a reviewer to read such an extensive comparison based paper without any discussion of correlation coefficients. At numerous points in the text (e.g., pg 10 line 1, page 12 line 7, and several others), authors not only compare sub-figures with each other, but sub-figures across different figures, all of which in my mind points to the inability of the figures in the main manuscript to convey the authors' message. This includes comparisons of $NO_2$ and $SO_2$ altitude patterns, remote versus in situ comparisons, and also comparison of patterns based on measured versus modelled wind data. I recommend using Spearman correlations instead of Pearson correlations because that coefficient better suits atmospheric measurement data (data at 0 and around 0 has value).

I strongly recommend the use of the correlation coefficients whereever possible, and a supporting figure in the supplement showing the relationship of the variables that are the

basis of the coefficients. Additionally, there are several places where the arguments the authors make (e.g., pg. 20 lines 14-18, pg. 22 lines 28-29, pg. 24 lines 13-15, pg 25 lines 5-6, Conclusion pg 30 Lines 18-20) could have easily been backed by correlations. This is also true for the relationships of surface/column ratio to BLH (Sect. 5), where the authors state positive or negative correlations without a coefficient.

4. **Choice of meteorological variables and focus on wind speed for horizontal transport**
The authors have focused Sect. 4 on a discussion of horizontal transport using wind speed and a later section on boundary layer height (Sect. 5). However, it is unclear to me that local wind speeds are representative of horizontal transport, and I believe that the only way to really model that transport is using 3D CTMs. The lack of references in Sect. 4 is also telling. Can the authors cite previous work that systematically shows local wind speeds and directions are representative of transport on the 10s of km scale? This would address the linkage of the wind speed discussion to plumes (e.g., pg 1 lines 27).

This critique above also made me question the broader theme of the paper: relationships of pollutant concentrations with specific meteorological variables. Given that they have access to detailed meteorological data, including measured temperature and several other modeled meteorological variables from ERA-5, did the authors consider conducting a PCA analysis of all meteorological data to see what are the key components that explain the data? If the variables presented here, wind direction, wind speed, and boundary layer height separate out as clear and separate components, this analysis makes sense. However, if other variables stand out such as temperature or precipitation, then the authors are making the age-old mistake of drawing conclusions from correlations when really, they want to get as close to causation as possible. And the best way to do that would be to identify the key PCA components of the underlying meteorology and then interpreting the data based on those components. At the least, the authors should consider the relationships of pollutant concentrations with altitude-resolved ventilation coefficient (see Fig. 6, Gani et al., ACP 2019) that combines the effects of wind speed and boundary layer height since, like all meteorological variables, they actually affect pollutants interlinked and not separately.

5. **Presentation of data**
There are several critical issues with the current style of presentation that need to be addressed.

- The authors have analyzed the data before presenting it. If anything, Section 6 should be Section 3, the first results section. In this current format, the authors jump straight into comparing remote sensing and insitu data using a complicated figure, that combines wind direction, concentration, and altitude, which is a difficult and unusual transition.

- The authors have conducted so many different analyses with different purpose that it is hard to keep track of all of them. I suggest adding a table at the end of the methods section listing the relevant pollutant, method used for VCD measurement, method used for in situ measurement, and if not, model used as replacement. This would allow you to cut a lot of text used for describing each analysis pipeline.

- Unnecessary length of manuscript: the manuscript is unnecessarily long, and large sections of the paper add little to the manuscript. For example, most sub-figures in Sect. 5 do not add much to the manuscript and could be placed in the Supplement. Also, Figs. 3-5 could be styled along the lines of Fig. 9, and separate detailed figures could be placed in the supplement. Additionally, as discussed above, sections 4 and 5 have relatively weak foundations, and should be shortened significantly, or large parts (e.g, Figs. 6-8) moved to the supplement. Only focus on the core messages for these sections instead of describing them in much detail.

- Several method descriptions are in other sections (e.g., pg 7 line 16-17, pg 10 lines 14-16, pg. 14 lines 6-10, pg. 20 lines 10-11, 18-20). This made reading the manuscript difficult. The authors should bring together all such descriptions back to the methods section.

- Before Fig. 10, the authors based all arguments on the surface to column ratio, but after, switch to column to surface. This is unnecessary, and should be kept consistent.

6. **Logical extrapolation and lack of explanation for multiple phenomena**

The authors have presented several phenomena (e.g., pg 11 lines 16-18, pg. 19 lines 1-2, pg. 28 lines 2-3) that they have not explained or explicitly figured out ways to address or are themselves perplexed by. I suggest going back to the literature to identify such instances happening elsewhere, or find explanations for why its happening here. One such instance seems fine to accommodate but multiple such instances in the same manuscript make it seem like the authors are simply putting data together and not interpreting it.

**Minor comments**

1. Pg. 3 Last sentence on lines 19-20 need a citation

2. Pg. 4 Last sentence on lines 16-18 should be moved to Sect. 2.5 and a reference to Sect. 2.5 should be made here

3. Pg. 5 line 10 Does the Pratmo model use wind data and other meteorological data. Explain.

4. Pg. 5 Last sentence on lines 14-15 what does "comparable" refer to? Comparable to what? Detail the % here.

5. Pg. 5 lines 22-23 Define fine and coarse mode by size. Also, detail the uncertainty. See comment 2 for broader issues with uncertainty.

6. Pg. 6 line 16 What does "both" refer to? Name the two polarizations.

7. Pg. 6 line 19 Why was this specific wavelength used even though other wavelengths were mentioned?

8. Pg. 9 line 21 Fig 3d suggests its 340 deg and not 280 deg in the cold season

9. Altitude figures e.g., Fig 3 the authors could draw average BLH estimates on this figure itself

10. Pg. 11 lines 14-15 cite Figs. 3d-e with this sentence

11. Figs. 3c, 3f show data for all directions; if not in the main manuscript, then in the supplement. Also, pg 12 line 1 2c is likely 3c.

12. pg 13 line 15-pg 4 line 2 phenomenon happening for NO2 but not for SO2. Why?

13. pg. 14 line 16, pg. 22 lines 10-11 add figure reference

14. Make fig. 6 in steps of 100 m and put other Figs. in the SI

15. pg. 22 lines 4-5 I disagree with the authors' assessment re: "a general agreement between the results based on WindRASS (see circles symbols) and reanalysis (see square symbols) data…" There are visibly influential deviations in 4 out of 6 subfigures.

16. Fig. 10 need to show 280 deg as well. Could add in the SI

**References**

1. Zhao et al., ACP 2019 https://doi.org/10.5194/acp-19-10619-2019
2. Gani et al., ACP 2019 https://acp.copernicus.org/articles/19/6843/2019/

---

## Author Response (AR1)

We thank Referee #1 for the suggestions and comments on this work. The paper has been revised based on the referee's comments (in red). Detailed answers have been provided (in blue).

This study compares the difference between in situ and remote sensing observations for air pollutants in from different perspectives over Canadian Oil Sands. The manuscript is pretty informative. However, the key contribution and findings are not clear, the large number of acronyms are difficult to follow, and the text is really long and it is suggested to keep it succinct.

Done. We shortened the text by moving Section 4 to the Appendix, and reduced the number of acronyms.

Detailed comments are given below:

Abstract

Line starting with 'Compared to an in situ instrument that provides…', that applies to nadir sensors, limb and occultation sensors provide vertical profiles.

Done.

*Compared to an in situ instrument that provides air quality conditions at the ground level, most remote sensing instruments (nadir viewing) are sensitive to a broad range of altitudes, often providing only integrated column observations.*

Line starting with 'Elevated SO2 VCDs are clearly observed…', are observed or were observed?

Done.

*Elevated $SO_2$ VCDs were clearly observed for times with south and south-eastern winds, particularly at 200–300 m altitude (above ground level).*

Do not mix-use British or American English, like analyzed and modelled, keep consistency

Thanks! This has been addressed with our best efforts.

Line starting with 'In addition to measured wind…' Maybe not all readers across the world know what ERA-5 reanalysis is, it is a good practice to give its full form at the first appearance.

Done.

*In addition to measured wind data and lidar observed boundary layer height (BLH), modelled wind profiles and BLH from ECMWF Reanalysis v5 (ERA-5) have been used to further examine the correlation between column and surface observations.*

Line starting with 'The results show that...' The findings are not significantly strong or exciting to intrigue the readers.

We have rephrased it as suggested.

*The results show that the height of emission sources (e.g., emissions from high stacks or near surface) will determine the ratio of measured column and surface concentration values (i.e., could show positive or negative correlation with BLH). This effect will show impacts on the comparison between column observations (e.g., from the satellite or ground-based remote sensing instruments) with surface in situ measurements.*

Introduction

Maybe not 'even' in polluted urban areas, my guess is relatively high correlation is expected in un- or less polluted areas for NO2, SO2, except for O3.

We think the referee was referring to this sentence:

*Alternatively, a direct comparison of ground-based VCD observations with surface concentrations does not typically produce high correlations  (Dieudonné et al., 2013).*

For un- or less polluted areas, the satellite column observations are more challenging, due to a worse signal-to-noise ratio. So, we fully agree with the referee that this issue is not only for urban but almost "everywhere". So, we deleted "even ...".

Authors may want to use acronyms less to help readers follow the storyline clearly instead of wondering the meaning of individual acronyms.

Thanks for the suggestions. We have reduced the number of acronyms, such as "WBEA", "AOSR", "DIAL", "U340", "ECCC", "PGN", etc.

Ln 15, we compare or compared?

Done.

*Besides trace gas pollutants, we also compared and studied the differences between surface PM₂.₅ observations with remote sensing AOD data. PM₂.₅ concentration is also one of the three indicators in the Canadian Air Quality Health Index (Stieb et al., 2008).*

Ln 20, the cause of what?

Done.

*Utilizing information on detailed vertical wind field profiles and BLH, this work illustrates the cause of  the differences between ground-based remote sensing and in situ observations, as well as studies the ratios of the surface to column values for various wind directions and BLH values.*

Ln 20 again, what is the major difference of definitions on ground-level and in situ observations?

The ground-based remote sensing observations provide the total column of pollutants, whereas the in situ observations provide pollutant concentrations on the ground. We have deleted the sentence and replaced it with the following paragraph.

*In short, this study is focused on the difference between total column (measured using ground-based remote sensing technique) and surface/ground-level concentrations (by in situ observations) of air pollutants. Measurements of wind profiles and BLH were used to examine their impact on that difference. The possibility of using modern reanalysis-modelled data instead of direct measurements of wind profiles and BLH were also explored. *

Ln 25, this bit 'integration period differences' comes out of sudden, readers may want to know why do we care about the integration period.

Thanks! We have rephrased this part.

*In Sect. 3, the  observation conditions (mainly weather) induced differences are evaluated.*

Section 2

It would be lovely to have a map of the study area that makes the paper more illustrative.

Figure 2 in the manuscript was the map (also plotted with satellite SO2 and NO2 VCD data). To make it clear, we have moved it to this section.

*Fort McKay is a small town (population of 600) surrounded by seven oil sands surface and two in-situ mining facilities to the north and south. Satellite maps showing the observation site and surrounding oil sands areas are shown in Fig. 1.*

Ln 5 What would the inspiration of this study be as the Fort McKay is a very small town? How would people be informed and apply the findings here to other polluted regions, as most heavy pollution that threatens people's life occur at densely populated urban areas.

We thank the referee for this insightful thought. As already provided in Section 2, one of the major outcomes of this work is that we found the pollution level at Fort McKay is largely dependent on the wind direction (and some other meteorological factors, such BLH).

*Thus, the pollution level at  the site is largely dependent on the wind direction. Therefore, the planning of communities close to industrial activities should consider regional climatology factors, such as prevailing wind directions.*

Ln 5 I do suggest the authors use acronyms less to make the life of readers easier.

Done. For example, we removed acronyms for the Wood Buffalo Environmental Association (WBEA), Nd: YAG, U340, AOSR, etc.

Section 2.1

Just curious, would it be considered as a 'recently developed' instrument as it has at least ten years of history?

Done.

*The Pandora spectrometer is a  ground-based remote sensing instrument that measures solar and sky spectral radiation in the UV and visible part of the spectrum (Herman et al., 2009; Szykman et al., 2019).*

Ln 30 What is U340 bandpass filter? What is ECCC? What is PGN?

Done. We agree with the referee there were too many unnecessary acronyms and technical details.

*The Pandora instrument consists of an optical head sensor, mounted on a computer-controlled sun-tracker, and connected to a commercial Avantes array spectrometer by means of an optical fibre. To allow for the detection of different absorbers, the instrument periodically measures UV spectra with the U340 bandpass filter with a cut-off limit at 380 nm on and off, with an interval of about 90 seconds. The 306–330 nm spectral interval was used for $SO_2$ spectral retrievals (ECCC research retrieval) and the 400 to 440 nm interval was used to retrieve $NO_2$ (PGN official retrieval, version nvs1p1-7) (Fioletov et al., 2016; Zhao et al., 2020).*

Ln 5 no. 104, does this serial number really matter?

Yes. There were two Pandora spectrometers been deployed at the site (no. 104 and 122), as described in this paragraph. The PGN and ECCC teams spend efforts in making sure the bias between the two systems has been corrected and accounted for. I.e., to have a uniform time series from 2014 to 2019. Also, when Pandora no. 122 was used, it also performed multi-axis observations (not included in this work), which reduced its direct sun (total column) sampling frequency. We could not provide too much technical details here, but still want to inform the reader of these important instrumental changes.

Ln 10 '…is given by (Herman et al., 2009)' is not a good format for reference.

Done.

*A detailed description of the Pandora spectrometer and its total column $NO_2$ retrieval algorithm is given by* Herman et al. (2009).

Ln 10 How does this Pratmo box model work and why it can remove NO2 in the stratosphere? Sometimes it's not the case the more information the merrier.

Done.

*In order to isolate tropospheric $NO_2$ VCD from the total column VCD measured by Pandora, stratospheric $NO_2$ partial columns were subtracted from Pandora measurements (following the method described in Zhao et al., 2019).*

Ln 15 What does the retrieved SO2 refer to? Is it from OMI? If it's from a project, we need a reference to it.

Done. The $SO_2$ column data is also retrieved from Pandora.

*For $SO_2$ data, as the only sources are near the surface (as no comparable $SO_2$ quantities were in the stratosphere during the analyzed period), the retrieved total column $SO_2$ ($SO_2$ VCD from Pandora observations) are directly been used in this study.*

Section 2.4

What is Q-switch Nd:YAG?

Done. Neodymium-doped Yttrium Aluminum Garnet (Nd:YAG) laser. To avoid more unnecessary details, we only provided the reference for the system.

*The lidar simultaneously emits two wavelengths laser light (1064 and 532 nm, Neodymium-doped Yttrium Aluminum Garnet laser, see references in Strawbridge, 2013) at energies of approximately 150 mJ pulse$^{-1}$ wavelength$^{-1}$ and detects the backscatter signal at 1064 nm and both polarizations at 532 nm.*

Section 3

What are coincident observations? Do the authors mean overlapped observations?

Yes.

*Thus, only coincident (overlapped) observations from both remote sensing and in situ instruments were included in the analysis.*

P9 Ln 5 What are selected wind directions? How did the authors do the selection, do you mean upwind or prevalent direction?

Done. It was described in the caption "*The white dashed lines show the centre of the wind sectors.*". Since this figure is now Fig. 1, we modified its caption accordingly.

*Figure 1. Satellite maps (© Google Maps) of the Athabasca Oil Sand Region  masked with satellite observations . The Pandora spectrometer, sunphotometer, WindRASS, lidar, and in situ instrument were located at the observation site represented by a white circle. The two largest upgraders in the mining areas are shown by red triangles. The white dashed lines show the centre of the wind sectors. Maps are masked with pixel averaging of total column $SO_2$ and tropospheric column $NO_2$ (2018–2021) from TROPOMI satellite instrument (McLinden et al., 2020).*

Conclusion

It would be better to have a concise conclusion

Done. We revised the conclusion to be more concise.

*The magnitude of the $SO_2$ VCD and surface concentrations reach their maximum for winds from 160° ± 30° directions at 200–300 m altitudes .*

*As a result, NO$_2$ VCD shows a more uniform sensitivity to winds from near surface to up to 300 m (peak value at 260–290 m altitude), while in situ measured NO$_2$ surface concentrations show a strong sensitivity to near-surface winds (peak value at 60–70 m altitude).*

~~The horizontal transport sampling differences show that local sources have the largest impact on observations when the wind speed is low and the pollutants are not transported far from the source (as in the case of some NO$_2$ emissions). For elevated sources (e.g., SO$_2$ emitted from high stacks), the moderate wind is substantial to bring the pollutant to the measurement site. Compared to in situ instruments, remote sensing observations are more sensitive to higher wind speed conditions, i.e., transported pollutants.~~

 *Here we show that linking these two measurements could be even more complicated, as they have more sampling differences than observations of trace gases.*

*Further results show replacing measured wind profiles and BLH by ERA-5 data could also preserve these features. Thus,*  *these ERA-5 data also can be utilized to reveal pollutants' vertical distribution and mixing conditions, which can be used as critical information when converting remote sensing column data to surface values.*

This analysis of surface to column ratios also shows that the column values cannot be converted to surface by just one value of the ratio. Depending on the wind direction *(and season),* the ratio for directions related to the pollution sources could be a factor of two larger than these from "clean" directions.

We thank Referee #2 for the suggestions and comments on this work. The paper has been revised based on the referee's comments (in red). Detailed answers have been provided (in blue).

Reviewer comments for:

**The differences between remote sensing and in situ air pollutants measurements over the Canadian Oil Sands**

This review is for the above manuscript submitted for publication in Atmospheric Measurement Techniques. The manuscript focuses on two key intercomparisons for data collected at or near Canadian Oil Sands: 1. in situ instrumentation measuring surface-level concentrations of multiple criteria air pollutants (SO2, NO2, PM2.5) and remote sensing instrumentation measuring total column concentrations of the same pollutants, and 2. Vertically resolved wind profiles measured at site and modelled wind profiles obtained from an international dataset (ERA-5). The authors conduct these comparisons primarily by linking altitude-based wind direction with concentration data. The authors find the presence of higher concentrations linked to certain wind directions irrespective of altitude for all three pollutants, and attribute these higher concentrations to key local sources in the region. Further, the authors also note the dominance of higher concentrations linked to certain altitudes (while fixing the wind direction), and present it as evidence of a strong influence of winds at that altitude on the measurement. The authors also repeat the analysis above with modelled wind data, and find similar results, suggesting that in absence of measured wind data, modelled data can be used. The authors also note the differing temporal coverage of data from in situ surface and remote sensing instrumentation, and quantify the biases in sampled and actual variations in BLH and pollutant concentrations. While the presentation of the altitude-concentration plots resolved by wind direction were very interesting, and the findings seem plausible too, I think that this manuscript lacks an underlying methological framework needed to conduct these comparisons.

Additionally, the current choice of presentation is very difficult for the reader to digest, and requires substantial changes as well. I suggest the authors **revise and resubmit** this manuscript for the following major issues.

1. **Issues with motivation of the paper.**

The authors chose multiple lines to motivate this work. This includes "applying satellite measurements to surface air quality applications" such as "air pollution monitoring" and "linking satellite air quality observations and surface in situ measurements". However, it is unclear how the authors have addressed these specific knowledge gaps in the manuscript. I recommend that the authors list clear objectives in the introduction and address those objectives step by step in different sections. This would also affect the abstract and the conclusions section, which would become more streamlined.

Additionally, the authors' literature review provides little motivation to compare in situ and ground based remote sensing data, with previous data showing bias <10% (Zhao et al., ACP 2019). If anything, the authors seem to have shown that in situ measurements combined with modelled ERA-5 data works really well, and that is a substantial finding. However, some of the other minutae can be removed from the conclusions then, with only key message kept in there.

We agree with the referee on this and more detailed objectives are provided in the revised manuscript.

In short, this study is focused on the difference between total column (measured using ground-based remote sensing technique) and surface/ground-level concentrations (by in situ observations) of air pollutants. Measurements of wind profiles and BLH were used to examine their impact on that difference. The possibility of using modern reanalysis-modelled data instead of direct measurements of wind profiles and BLH were also explored.

We appreciate the thoughts and suggestions from the referee. Regarding some "good data agreement" (-7%) between in situ and ground-based remote sensing, i.e., Zhao et al. (ACP 2019), such value is the mean bias for a large group of observations in various conditions. However, for some very challenging conditions (such as low BLH conditions, e.g., morning hours), the bias between the two types of instruments will be larger. In short, currently, the remote sensing community is still working on improving its capacity to retrieve high-quality surface concentrations. We believe such a "7% bias" is still too high for air quality monitoring. This is why the research community is working on this, and (in this work) we are trying to "understand" where such a bias came from and how we can improve it further.

**2. Lack of uncertainty analysis**

In a paper like this with substantial use of modelled data, I think a detailed subsection on uncertainties of the different pollutants modelled or measured using different instruments is needed in the Methods section. I suggest that the authors also add a small table of the pollutants/meteorological variables measured/modelled, the instruments used to measure/model them, and their average concentrations, standard deviations, and uncertainties. Also, for atmospheric pollutant data, geometric mean and geometric standard deviation may be more representative (please check). For wind direction, median wind direction (and not mean) would be more representative. Similarly, all instances of mentioning specific concentration or DU values in the manuscript should be accompanied with associated uncertainty (e.g., on pg 10 line 4).

Done. We added a table to provide these statistics for measured pollutants.

*In addition to wind data, the boundary layer height from ERA-5 has also been used to examine the correlation between column and surface observations. The statistics of measured pollutants are*

*summarized in Table 1.*

**Table 1. Statistics of measured pollutants.**

| Pollutants | Measurement sources for comparisons | Mean (median) | Standard deviation | Uncertainties |
|---|---|---|---|---|
| Sulphur Dioxide (SO$_2$) | Pandora (column in DU) | 0.22 (0.10) | 0.54 | 0.05 |
| | In situ (surface concentration in ppbv) | 1.09 (0.35) | 3.15 | 1 |
| Nitrogen Dioxide (NO$_2$) | Pandora (column in DU) | 0.16 (0.14) | 0.15 | 0.001 |
| | In situ (surface concentration in ppbv) | 7.40 (4.42) | 7.98 | 0.4 |
| Aerosol | Sunphotometer (AOD) | 0.10 (0.06) | 0.19 | 0.02 |
| | In situ PM$_{2.5}$ (surface concentration in µg m$^{-3}$) | 8.59 (5.16) | 37.75 | 0.5 |

We fully agree with the referee that the in situ air pollution measurements could use geometric mean and std to represent their statistics. The geometric mean is appropriate in cases of lognormally-distributed variables. It is indeed often the case for surface concentrations of air pollutants: the values are typically low, but there are some cases of high values when a plume passes over the measurement site. However, our case is different since our site is located in a polluted area. For NO$_2$, the surface concentrations fluctuate in the range of 20-40 ppbv with no obvious spikes (see Figure 2 in the revised paper). For SO$_2$, the values are typically low and with low variability, except for one particular wind direction (from the emission source). Thus, our analysis is done for data grouped by the wind direction. For AOD/PM$_{2.5}$, the distribution indeed is close to lognormal due to the impacts of forest fires, but we decided to use the same approach as for NO$_2$ and SO$_2$ for consistency. Histograms of these pollutants and their log-transformed data are shown below (Figs. R1 and R2).

[Figure]

**Figure R1. Histograms of air pollutants.**

**Figure R2. Log-transformed histograms of air pollutants.**

**3. Comparison-based research without correlation coefficients?**

It was terribly inconvenient as a reviewer to read such an extensive comparison based paper

without any discussion of correlation coefficients. At numerous points in the text (e.g., pg 10 line 1, page 12 line 7, and several others), authors not only compare sub-figures with each other, but sub-figures across different figures, all of which in my mind points to the inability of the figures in the main manuscript to convey the authors' message. This includes comparisons of NO2 and SO2 altitude patterns, remote versus in situ comparisons, and also comparison of patterns based on measured versus modelled wind data. I recommend using Spearman correlations instead of Pearson correlations because that coefficient better suits atmospheric measurement data (data at 0 and around 0 has value).

I strongly recommend the use of the correlation coefficients whereever possible, and a supporting figure in the supplement showing the relationship of the variables that are the basis of the coefficients. Additionally, there are several places where the arguments the authors make (e.g., pg. 20 lines 14-18, pg. 22 lines 28-29, pg. 24 lines 13-15, pg 25 lines 5-6, Conclusion pg 30 Lines 18-20) could have easily been backed by correlations. This is also true for the relationships of surface/column ratio to BLH (Sect. 5), where the authors state positive or negative correlations without a coefficient.

We thank the referee for the suggestion to use correlation coefficients in comparisons. However, correlation coefficients can be misleading in some cases. For example, the reviewer mentioned Page 25 where Figure 12 was discussed (in the original manuscript on AMTD). Our point was that mean daily variation patterns are similar for in situ and VCD values. In the case of $SO_2$ (Fig. 12b), there are no obvious diurnal variation patterns (except for early morning), so the correlation between the in situ and remote sensing diurnal pattern is only -0.3621. The table below shows all correlation coefficients calculated for Figure 12.

**Table R1. Correlation coefficients between Remote sensing and in situ observed $SO_2$, $NO_2$, and aerosol data (AOD and $PM_{2.5}$) for Figure 12.**

| Panel | Pearson | Spearman |
|-------|---------|----------|
| a | -0.59 | -0.53 |
| b | -0.36 | -0.13 |
| c | 0.82 | 0.56 |
| d | 0.82 | 0.88 |
| e | 0.22 | 0.36 |
| f | 0.45 | 0.45 |

On the other hand, we also added a few correlation coefficient estimates where appropriate.

*The results clearly show that, for the $SO_2$ plume from upgraders (see Fig. 10c), the $SO_2$ column to surface ratio is*  *a monotonic declining function of BLH (with a Spearman correlation coefficient of -0.83).*

*In contrast, a similar analysis for $NO_2$ observations shows different features (see Figs. 10d-f). This time, as $NO_2$ has two major source directions (40° and 160° directions), Figs. 10e and f show clear and similar column to surface ratio patterns, i.e., this ratio is positively correlated with BLH (with a Spearman correlation coefficient of 0.94 and 1).*

*However, in cold seasons, there is a positive correlation between the $AOD/PM_{2.5}$ ratio to BLH data from 40° direction (see Figs. 11h; with a Spearman correlation coefficient of 1). Thus, in cold seasons, from this direction, the aerosol loads are closer to the surface than vertically mixed. A table of detailed correlation coefficient values for Figs. 10 and 11 are provided in Appendix B.*

*The results show ERA-5 data could also assist the work in terms of separated pollutants' signal from different sources (see Figs. B2, B3, and Table B1).*

In Appendix B, we included a new table (a simplified version of Table R2 below, without Pearson R) to provide detailed correlation coefficient estimates for Figs. 10, 11, C2, and C3.

*Table R2. Correlation coefficients between median values of column to surface ratio and BLH height for Figure 10.*

| Panel | Fig. 10 | | Fig. 11 | | Fig. C2 | | Fig. C3 | |
|---|---|---|---|---|---|---|---|---|
| | Pearson | Spearman | Pearson | Spearman | Pearson | Spearman | Pearson | Spearman |
| a | -0.65 | -0.49 | 0.51 | 0.37 | -0.41 | -0.43 | 0.67 | 0.60 |
| b | -0.30 | -0.09 | -0.16 | -0.03 | -0.89 | -0.83 | -0.21 | -0.26 |
| c | -0.75 | -0.83 | -0.41 | -0.54 | -0.74 | -0.94 | -0.57 | -0.09 |
| d | 0.91 | 0.94 | 0.94 | 0.94 | 0.89 | 0.83 | 0.97 | 1.00 |
| e | 0.96 | 0.94 | 0.91 | 0.89 | 0.94 | 0.94 | 0.97 | 1.00 |
| f | 0.91 | 1.00 | 0.93 | 1.00 | 0.78 | 0.66 | 0.87 | 0.94 |
| g | 0.94 | 0.94 | 0.97 | 0.94 | 0.94 | 0.94 | 0.98 | 1.00 |
| h | 0.20 | 0.20 | 0.96 | 1.00 | -0.42 | -0.31 | 0.75 | 1.00 |
| i | 0.68 | 0.49 | 0.79 | 0.66 | 0.39 | 0.26 | 0.74 | 0.60 |

**4. Choice of meteorological variables and focus on wind speed for horizontal transport**

The authors have focused Sect. 4 on a discussion of horizontal transport using wind speed and a later section on boundary layer height (Sect. 5). However, it is unclear to me that local wind speeds are representative of horizontal transport, and I believe that the only way to really model that transport is using 3D CTMs. The lack of references in Sect. 4 is also telling. Can the authors cite previous work that systematically shows local wind speeds and directions are representative of transport on the 10s of km scale? This would address the linkage of the wind speed discussion to plumes (e.g., pg 1 lines 27).

This critique above also made me question the broader theme of the paper: relationships of pollutant concentrations with specific meteorological variables. Given that they have access to detailed meteorological data, including measured temperature and several other modeled meteorological variables from ERA-5, did the authors consider conducting a PCA analysis of all meteorological data to see what are the key components that explain the data? If the variables presented here, wind direction, wind speed, and boundary layer height separate out as clear and separate components, this analysis makes sense. However, if other variables

stand out such as temperature or precipitation, then the authors are making the age-old mistake of drawing conclusions from correlations when really, they want to get as close to causation as possible. And the best way to do that would be to identify the key PCA components of the underlying meteorology and then interpreting the data based on those components. At the least, the authors should consider the relationships of pollutant concentrations with altitude-resolved ventilation coefficient (see Fig. 6, Gani et al., ACP 2019) that combines the effects of wind speed and boundary layer height since, like all meteorological variables, they actually affect pollutants interlinked and not separately.

We thank the referee for the suggestion that comparing the measured meteorological variables with detailed 3D CTM, which could reveal more details such as key components that explain the data. However, ERA-5 data that has been used here is not a good option. ERA-5 hourly data has a 0.25° × 0.25° horizontal resolution (about 30 km × 15 km in this region). Note that, as described in Section 2, the distance between the observation site to the two $SO_2$ sources are 16 km and 23 km, respectively. Thus, the reanalysis data's spatial and temporal resolution is simply not enough to study the "representative of horizontal transport" as suggested. Given the sources are too close, we would need dedicated CTM simulations to have horizontal resolution at least on a 5 km scale, which is unfortunately not available. We also agree with the referee that we should include more discussion about meteorology and its effects. We included winds (vertical and horizontal) and BLH data, as these were the factors that have well-known effects on air pollution transports and mixing (e.g., Fioletov et al., 2017; Zhao et al., 2020). Other suggested variables, such as temperature or precipitation were not included. First, BLH is also greatly affected by temperature, but it is a more straightforward measure to indicate the vertical mixing of air pollutants (e.g., Kollonige et al., 2017). As described in Section 6, the direct-sun remote sensing instruments (Pandora and sunphotometer) only work in sunny conditions (and we studied coincident observations only). Thus, precipitation would not be a factor to be considered.

Also, the pollutants we measured are mainly controlled by the nearby point source emissions (from the Syncrude Mildred Lake plant and the Suncor Millenium Plant). Thus, the suggested ventilation coefficient could not resolve the issues. The ventilation coefficient is described in Gani et al., ACP 2019 as "PBL × wind speed", which is sometimes referred to as the normalized dilution rate. However, we must point out that this simplified "dilution rate" could work only if the pollutants are more or less uniformly mixed in a region. I.e., for our case, if the measurement site is not at the downwind of those point sources, this ventilation coefficient will not affect the observations at all (e.g., see Fig. 6).

We still did the analysis for this combined meteorological parameter. As expected, a clear negative correlation between column to surface ratio and ventilation coefficient can only be found for wind directions from 160±30 degrees for $SO_2$ (see figure below).

[Figure]

**Figure R3. $SO_2$ column to surface ratio binned by ventilation coefficient (BLH×wind speed).**

Similarly, for $NO_2$, the ventilation coefficient also shows a similar feature as the direct use of BLH for polluted directions (see figure below; note that 280 ±30 degrees are "clean air" directions).

[Figure]

**Figure R4. NO₂ column to surface ratio binned by ventilation coefficient (BLH×wind speed).**

Last, the analysis for aerosol is shown below, which did not provide any suggestions on whether this combined meteorological factor could play a better role than directly using BLH.

[Figure]

**Figure R5. Aerosol column to surface ratio binned by ventilation coefficient (BLH×wind speed).**

In short, we investigate the suggested parameter. However, due to the special conditions for the current site, we could not find a good reason to include this combined meteorological factor. We also plotted a similar figure as suggested (same as Fig. 6 in Gani et al., ACP 2019). However, we could not see similar features from the current site (see figure below). Thus, we

decided not to include the suggested ventilation coefficient in this work.

[Figure]

**Figure R6. PM₂.₅ vs. ventilation coefficient, colour-coded by density of data points.**

Last, we included some of this information at the end of Section 5.

*The results show ERA-5 data could also assist the work in terms of separated pollutants' signal from different sources (see Figs. B2, B3, and Table B1). In general, besides wind conditions (directions and speed) and BLH, there could be other meteorological factors played roles in describe the difference between in situ and remote sensing measurements. We also examined combined meteorological factor, such as the ventilation coefficient (BLH × wind speed), which is also referred to as the normalized dilution rate (e.g., Gani et al., 2019). However, no improvement compared to BLH-based results was found, likely due to the complexity of the pollution source distribution. More detailed high-resolution modelling work is needed to further understand the meteorological impacts to local observations..*

**5. Presentation of data**
There are several critical issues with the current style of presentation that need to be

addressed.

- The authors have analyzed the data before presenting it. If anything, Section 6 should be Section 3, the first results section. In this current format, the authors jump straight into comparing remote sensing and insitu data using a complicated figure, that combines wind direction, concentration, and altitude, which is a difficult and unusual transition.

- The authors have conducted so many different analyses with different purpose that it is hard to keep track of all of them. I suggest adding a table at the end of the methods section listing the relevant pollutant, method used for VCD measurement, method used for in situ measurement, and if not, model used as replacement. This would allow you to cut a lot of text used for describing each analysis pipeline.

- Unnecessary length of manuscript: the manuscript is unnecessarily long, and large sections of the paper add little to the manuscript. For example, most sub-figures in Sect. 5 do not add much to the manuscript and could be placed in the Supplement. Also, Figs. 3-5 could be styled along the lines of Fig. 9, and separate detailed figures could be placed in the supplement. Additionally, as discussed above, sections 4 and 5 have relatively weak foundations, and should be shortened significantly, or large parts (e.g, Figs. 6-8) moved to the supplement. Only focus on the core messages for these sections instead of describing them in much detail.

- Several method descriptions are in other sections (e.g., pg 7 line 16-17, pg 10 lines 1416, pg. 14 lines 6-10, pg. 20 lines 10-11, 18-20). This made reading the manuscript difficult. The authors should bring together all such descriptions back to the methods section.

- Before Fig. 10, the authors based all arguments on the surface to column ratio, but after, switch to column to surface. This is unnecessary, and should be kept consistent.

Following the suggestions, we have moved Section 7 to Section 3, and moved Section 4

(Figures 6-8) to Appendix. A table to show measurement types is provided in Section 2. As the complication of this work, some detailed method descriptions of individual measurements should still be kept at their best possible position, i.e., where we need to provide such detailed information.

For Fig. 10, the shift from surface-to-column ratio to column-to-surface ratio has a reason. We have plots of surface-to-column ratio that will show more "prominent" features for high BLH conditions (which is not ideal in this case). I.e., when BLH is low (0-0.2 km), of course, the surface-to-column ratio is very small for elevated $SO_2$ plume. Figure R7 is provided below for the referee. Thus, to better illustrate the effect of emission height, we decided to use the column-to-surface ratio.

[Figure]

**Figure R7. Surface to column ratio vs. BLH.**

Note that, this reason was provided in the original manuscript.

*To better illustrate the effect of low BLH conditions on the $SO_2$ plume's vertical distribution, in this section, we plotted the column to surface ratio.*

6. **Logical extrapolation and lack of explanation for multiple phenomena**

The authors have presented several phenomena (e.g., pg 11 lines 16-18, pg. 19 lines 1-2, pg. 28 lines 2-3) that they have not explained or explicitly figured out ways to address or are themselves perplexed by. I suggest going back to the literature to identify such instances happening elsewhere, or find explanations for why its happening here. One such instance seems fine to accommodate but multiple such instances in the same manuscript make it seem like the authors are simply putting data together and not interpreting it.

We thank the referee for the suggestions. However, the discrepancy between remote sensing and in situ measurements is a well-known issue in this research community. From what we understand, this is the first work trying to provide solid observation-based (quantified) explanations for some of these differences. I.e., we are not just attributing the differences between instruments to simple words, such as "sampling difference". We agree with the referee that we still cannot explain all the phenomena been observed, but we hope these efforts will cast some light on the path to resolving these scientific issues and questions.

**Minor comments**

1. Pg. 3 Last sentence on lines 19-20 need a citation

Done.

*Remote sensing observations of aerosols face more challenges than ozone and $NO_2$ (e.g., Herman et al.,*

*2009; Jeong et al., 2020).*

2. Pg. 4 Last sentence on lines 16-18 should be moved to Sect. 2.5 and a reference to Sect. 2.5 should be made here

Done.

3. Pg. 5 line 10 Does the Pratmo model use wind data and other meteorological data. Explain.

Based on a suggestion from Referee 1, we removed the information of Pratmo, but provided a reference to it.

*In order to isolate tropospheric $NO_2$ VCD from the total column VCD measured by Pandora, stratospheric $NO_2$ partial columns were subtracted from Pandora measurements ( following the method described in Zhao et al., 2019).*

4. Pg. 5 Last sentence on lines 14-15 what does "comparable" refer to? Comparable to what? Detail the % here.

We included more detailed explanations for this sentence. Stratospheric SO2 could be comparable to tropospheric SO2 in events such as volcanic eruptions. The typical background level of $SO_2$ in stratospheric is around 0.01 to 0.001 DU, i.e., less than 3% of the total $SO_2$ column.

*For $SO_2$ data, as the only sources are near the surface (as no comparable $SO_2$ quantities were in the stratosphere during the analyzed period, i.e., no $SO_2$ injection from volcanic eruptions), the retrieved total column $SO_2$ ($SO_2$ VCD from Pandora observations) are directly been used in this study.*

5. Pg. 5 lines 22-23 Define fine and coarse mode by size. Also, detail the uncertainty. See comment 2 for broader issues with uncertainty.

The size of fine and coarse modes were provided in the manuscript. Following the suggestion, the uncertainties of these measurements are now included in Table 1 in the revised manuscript. *Fine and coarse modes are essentially comprised of sub-micron and super-micron particle radii, respectively (O'Neill et al., 2001).*

6. Pg. 6 line 16 What does "both" refer to? Name the two polarizations.

This sentence has been rephrased.

*The lidar simultaneously emits two wavelengths laser light (1064 and 532 nm, Q-switch Neodymium-doped Yttrium Aluminum Garnet* *, see references in Strawbridge, 2013) at energies of approximately 150 mJ pulse$^{-1}$ wavelength$^{-1}$ and detects the backscatter signal at 1064 nm and 532 nm (and polarizations at 532 nm).*

7. Pg. 6 line 19 Why was this specific wavelength used even though other wavelengths were mentioned?
The algorithm to derive BLH data uses 532 nm measurements, which have good balance between signal strength and atmospheric penetration. Other wavelengths were mentioned to illustrate the capacities of the lidar system. Although we did not use results such as aerosol profiles, they can be used in future comparison and validation work to continue the current efforts.

8. Pg. 9 line 21 Fig 3d suggests its 340 deg and not 280 deg in the cold season

For Fig 3d (original manuscript), results from 280 and 340 degrees are all similarly low. Note that, we only can divide the wind directions into these coarse 60 degrees bins, and 280 and 340

degrees bins are close. In addition, the data from winter time is more sparse (e.g., see the wider shading areas represent the 1-sigma level of the measurements). As a result, we do not think the small differences in cold seasons mean 340 degrees is "more cleaner" than 280 degrees. Also, we still prefer to call 280 degrees "clean air" directions, also it has no major $NO_2$ sources (see the satellite maps below).

[Figure]

*Figure 1. Satellite maps (© Google Maps) of the Athabasca Oil Sand Region (AOSR) masked with satellite observations. The Pandora spectrometer, sunphotometer, WindRASS, lidar, and in situ instrument were located at the observation site represented by a white circle. The two largest upgraders in the mining areas are shown by red triangles. The white dashed lines show the centre of the wind sectors. Maps are masked with pixel averaging of total column $SO_2$ and tropospheric column $NO_2$ (2018–2021) from TROPOMI satellite instrument (McLinden et al., 2020).*

9. Altitude figures e.g., Fig 3 the authors could draw average BLH estimates on this figure itself

Following the suggestion, we did the calculation for BLH from ERA-5. For warm (May to Oct.)

and cold seasons (Nov. to Apr.), the median values of BLH are 390 and 208 m, respectively. For Fig. 3, due to the low success rate of WindRASS, we cut off the plot at 300 m. So, we could not plot these typical BLH heights on the suggested figures, but we provided such information in the related paragraph.

*The results may indicate that the vertical transport of $SO_2$ is more refined within the boundary layer due to a lack of vertical mixing in cold temperatures.* **Note that, for warm and cold seasons, the median values of BLH from ERA-5 are 390 m and 208 m, respectively.**

10. Pg. 11 lines 14-15 cite Figs. 3d-e with this sentence

Done.

*Compared to the graduate decreasing sensitivity above "plume height" in warm seasons (i.e., 200–300 m), both remote sensing and in situ observations show their sensitivity to $SO_2$ emissions decreased sharply after passing the plume height in cold seasons* **(Figs. 5d and e).**

11. Figs. 3c, 3f show data for all directions; if not in the main manuscript, then in the supplement. Also, pg 12 line 1 2c is likely 3c.

Thanks for the suggestion. But, we simplified these two sub-panels in the figure to only show key information, i.e., one from the polluted direction, and one from the clean air direction. This is to avoid overcrowded lines and too much unnecessary information.

12. pg 13 line 15-pg 4 line 2 phenomenon happening for NO2 but not for SO2. Why?

This is due to many factors, including different emission patterns of $NO_2$ and $SO_2$. Note that, $NO_2$ shows a much stronger seasonal variation than $SO_2$. It is also correlated with the emission source heights, in which $SO_2$ is mainly from high stacks above ground. All this information was

provided in Section 4.3 and conclusion.

*Comparing to the profiles of in situ $SO_2$ (elevated at 200–300 m; emissions from high stacks) and in situ $NO_2$ (two peaks, one near-surface and one elevated at 200–300 m; emissions from both near-surface mining fleet and high stacks), the sources of $PM_{2.5}$ from this direction could not easily be distinguished.*

*The magnitude of the $SO_2$ VCD and surface concentrations reach their maximum for winds from $160° \pm 30°$ directions at 200–300 m altitudes (which is about 38% and 27% higher than the value for winds from the same directions but of near surface winds, respectively). In contrast, the $NO_2$ emissions from $160° \pm 30°$ directions are from both high stacks and the mining fleets. As a result, $NO_2$ VCD shows a more uniform sensitivity to winds from near surface to up to 300 m (peak value at 260–290 m altitude), while in situ measured $NO_2$ surface concentrations show a strong sensitivity to near-surface winds (peak value at 60–70 m altitude). In cold seasons, the $NO_2$ surface to column ratio from $160°$ direction changes from 39 ppbv/DU at 70 m to 28 ppbv/DU at 200 m.*

13. pg. 14 line 16, pg. 22 lines 10-11 add figure reference

Done.

*Comparing to the profiles of in situ $SO_2$ (elevated at 200–300 m; emissions from high stacks) and in situ $NO_2$ (two peaks, one near-surface and one elevated at 200–300 m; emissions from both near-surface mining fleet and high stacks) (see Figs. 5 and 6), the sources of $PM_{2.5}$ from this direction could not easily be distinguished.*

14. Make fig. 6 in steps of 100 m and put other Figs. in the SI

We have moved the entire Section 4 to the Appendix.

15. pg. 22 lines 4-5 I disagree with the authors' assessment re: "a general agreement between the results based on WindRASS (see circles symbols) and reanalysis (see square symbols)

data..." There are visibly influential deviations in 4 out of 6 subfigures.

Here we are comparing coarse-modelled vertical results with detailed measurements. The feature we are looking for is the general agreement, i.e., if the modelled wind can be used to distinguish clean and polluted air's influence on the surface-to-column ratio. The measurements show clearly that such a ratio is higher from the polluted directions, while the model results presented the same pattern clearly for 4 out of 6 subfigures. Even in panels g and l the results from polluted directions are still slightly larger than clean air directions. Please note that, even for measurement results, panels g and l also show the smallest differences. We revised the description to make it clear.

*Regarding the surface to column ratios (0–300 m), the last two columns of Fig. 8 show a general agreement between the results based on WindRASS (see circles symbols) and reanalysis (see square symbols) data (i.e., the ratios are typically higher from polluted directions than clean air directions), although the altitudinal dependence has some differences.*

16. Fig. 10 need to show 280 deg as well. Could add in the SI

We had such information, but they will not add more new knowledge. I.e., for clean air directions, such a ratio will be very noisy (due to the nature of low signal-to-noise ratio for these directions). We still plotted them for the referee, as shown in Figures R3, R4, and R5 in this reply. We decided not to include them in the paper.

**References**

1. Zhao et al., ACP 2019 https://doi.org/10.5194/acp-19-10619-2019
2. Gani et al., ACP 2019 https://acp.copernicus.org/articles/19/6843/2019/